# The EIF3H-HAX1 axis increases RAF-MEK-ERK signaling activity to promote colorectal cancer progression

Huilin Jin[1,2,3,4,9], Xiaoling Huang[1,2,3,9], Qihao Pan[2,3,5,9], Ning Ma[1,2,3], Xiaoshan Xie[1,2,3], Yue Wei[1,2,3], Fenghai Yu[1,2,3], Weijie Wen[1,2,3], Boyu Zhang[1,2,3], Peng Zhang[1,2,3], Xijie Chen[1,2,3,6], Jie Wang[7], Ran-yi Liu [1,2,3,6]
[8], Junzhong Lin [8], Xiangqi Meng [1,2,3] ✉ & Mong-Hong Lee [1,2,3] ✉

Eukaryotic initiation translation factor 3 subunit h (EIF3H) plays critical roles in regulating translational initiation and predicts poor cancer prognosis, but the mechanism underlying EIF3H tumorigenesis remains to be further elucidated. Here, we report that EIF3H is overexpressed in colorectal cancer (CRC) and correlates with poor prognosis. Conditional *Eif3h* deletion suppresses color-ectal tumorigenesis in AOM/DSS model. Mechanistically, EIF3H functions as a deubiquitinase for HAX1 and stabilizes HAX1 via antagonizing βTrCP-mediated ubiquitination, which enhances the interaction between RAF1, MEK1 and ERK1, thereby potentiating phosphorylation of ERK1/2. In addition, activation of Wnt/β-catenin signaling induces EIF3H expression. EIF3H/HAX1 axis promotes CRC tumorigenesis and metastasis in mouse orthotopic cancer model. Sig-nificantly, combined targeting Wnt and RAF1-ERK1/2 signaling synergistically inhibits tumor growth in EIF3H-high patient-derived xenografts. These results uncover the important roles of EIF3H in mediating CRC progression through regulating HAX1 and RAF1-ERK1/2 signaling. EIF3H represents a promising therapeutic target and prognostic marker in CRC.

Colorectal cancer (CRC) is one of the most common causes of cancer-related deaths worldwide[1]. Major risk factors for CRC include seden-tary lifestyle, obesity, inflammatory bowel disease, and dysregulation of the gut microbiome[2–5]. CRC is mainly detected at advanced stages, and the prognosis is poor[6]. Although great efforts have been made in the past few decades, the definite molecular mechanisms of CRC development remain unclear.

The eukaryotic translation initiation factor 3, subunit h (EIF3H) is a subunit of eIF3 complex, which regulates processes of initiation and elongation during protein translation[7]. EIF3H can enhance protein synthesis and promote tumor progression, especially promote the translation of certain oncogenes[8–10]. Subunit EIF3H belongs to Jab1/Mov34/MPN+protease (JAMM) family, harbors consensus of MPN motifs that could have zinc-binding metalloprotease function[11]. In

[1]Department of General Surgery, The Sixth Affiliated Hospital, Sun Yat-sen University, Guangzhou, China. [2]Guangdong Provincial Key laboratory of Colorectal and Pelvic Floor Disease, The Sixth Affiliated Hospital, Sun Yat-sen University, Guangzhou, China. [3]Biomedical Innovation Center, The Sixth Affiliated Hospital, Sun Yat-sen University, Guangzhou, China. [4]Department of Hepatobiliary, Pancreatic and Splenic surgery, The Sixth Affiliated Hospital, Sun Yat-sen University, Guangzhou, China. [5]Department of Obstetrics and Gynecology, The Sixth Affiliated Hospital, Sun Yat-sen University, Guangzhou, China. [6]Department of Gastrointestinal Surgery, The Sixth Affiliated Hospital, Sun Yat-Sen University, Guangzhou, China. [7]Department of Radiation Oncology, Dalian Municipal Central Hospital, Dalian, China. [8]State Key Laboratory of Oncology in South China & Collaborative Innovation Center for Cancer Medicine, Sun Yat-sen University Cancer Center, Guangzhou, China. [9]These authors contributed equally: Huilin Jin, Xiaoling Huang, Qihao Pan.
✉e-mail: mengxq3@mail.sysu.edu.cn; limh33@mail.sysu.edu.cn

addition to the obligate function of translation initiation regulation, EIF3H has an alternative "moonlighting" function of deubiquitinase for its putative nonconserved MPN domain[12]. However, whether EIF3H plays a deubiquitinating role in CRC tumorigenesis and what are its specific substrates remain largely unexplored.

HAX1 (HS1 associated protein X-1) is a multifunctional protein involved in various pathophysiological processes and cellular metabolism such as apoptosis, calcium homeostasis, and cell migration[13–15]. HAX1 expression is elevated in multiple human cancers, including CRC, and CRC patients with HAX1 overexpression had a significantly poor overall survival[16]. However, the detailed mechanisms regulating HAX1 protein abundance and the functional role of HAX1 in CRC have not been well elucidated.

Here, we show that EIF3H expression is up-regulated in CRC. Conditional knockout of *Eif3h* in *villin*/intestinal epithelial cells leads to embryonic lethality. Heterozygous *Eif3h* intestinal epithelial cell knockout mice demonstrates haploinsufficiency in terms of attenuating CRC growth. Further, EIF3H promotes CRC tumorigenesis and metastasis via HAX1 activity in mouse model. Mechanistically, EIF3H functions as a deubiquitinase for HAX1 and stabilize it, which in turn enhanced the interaction between RAF1, MEK1 and ERK1, thereby potentiating phosphorylation of ERK1/2 and subsequent growth/metastasis promotion. In addition, activation of Wnt/β-catenin signaling transcriptionally induces EIF3H expression in CRC. Importantly, combination of Wnt inhibitor and RAF1/MEK inhibitor can suppress the tumor growth of patient-derived xenografts (PDXs) that have EIF3H/HAX1 overexpression. Our studies of the role of Wnt/EIF3H/HAX1 in enhancing RAF/MEK/ERK cascade reveal rational therapy for CRC intervention.

## Results

### EIF3H is overexpressed in CRC
To assess the clinical relevance of EIF3H, we investigated EIF3H expression in the CRC and normal tissue samples from TCGA data of patients with CRC, GSE41258, GSE8671 and GSE77434 datasets (Supplementary Fig. 1a, b). The results showed that EIF3H was significantly upregulated in the tumor samples. Further analyses showed that elevated EIF3H mRNA level was correlated with metastasis to lymph nodes and advanced AJCC stage (Supplementary Fig. 1c, d), implying EIF3H role in tumor progression. Upregulated EIF3H protein expression was observed in the Clinical Proteomic Tumor Analysis Consortium colon cancer database as well (Supplementary Fig. 1e). To validate the results from database, we detected EIF3H mRNA levels in 31 paired samples of CRC and adjacent normal tissues by qRT-PCR. EIF3H mRNA expression was significantly increased in tumors compared with normal tissues (Fig. 1a). Taken together, these data strongly indicated that EIF3H expression is associated with CRC development and prognosis.

### *Villin*–specific conditional *Eif3h* knockout attenuates colorectal tumorigenesis in AOM/DSS model
To further understand the functional role of epithelial EIF3H in the development of CRC in vivo, we generated conditional intestinal epithelial cells-specific *Eif3h*-deficient mice. Accordingly, we generated whole body *Eif3h*^flox/wt mice based on CRISPR/Cas9 system (Fig. 1b). Following breeding procedures, we generated inducible heterozygous *Eif3h* intestinal epithelial cell-specific knockout mice (*Eif3h*^flox/wt, *Villin*-CreERT) by crossing a mouse line containing floxed alleles of *Eif3h* (*Eif3h*^flox/flox) with another mouse line expressing Cre-recombinase under control of the *Villin* promoter (*Villin*-CreERT) (Fig. 1c). The genotypes of which were verified using PCR analysis (Supplementary Fig. 2a). Breeding did not result in progeny with homozygous deletion of *Eif3h* (*Eif3h*^flox/flox *Villin*-CreERT), indicating embryonic lethality of *Eif3h*^−/− mice (Supplementary Fig. 2a). There were no obvious physical, developmental abnormalities (body weight, colon length) found in *Eif3h*^flox/wt *Villin*CreERT mice (Supplementary Fig. 2b), thus they were used for further experiments.

As expected, after tamoxifen (TAM) induction, the colon epithelial cells from *Eif3h* heterozygous knockout mice (*Eif3h*^flox/wt *Villin*-CreERT+TAM, *Eif3h*^-/wt) group mice turned red which indicated the expression and function of CreERT, based on crossing with a reporter mouse line (B6-G/R, the intestinal epithelial cells will turn red from green when Cre enzyme functions). Meanwhile, further analyses of the gut of *Eif3h*^-/wt revealed a specific decrease of *Eif3h* staining in intestinal epithelial cells (Fig. 1d and Supplementary Fig. 2d, e) when compared with their *Eif3h*^flox/wt + TAM littermates (*Eif3h*^flox/wt). *Eif3h*^-/wt showed normal development with no overt phenotype in the gut (Supplementary Fig. 2c). *Eif3h*^-/wt mice and their *Eif3h*^flox/wt littermates were treated with azoxymethane (AOM) and dextran sodium sulphate (DSS) to induce colitis-associated colorectal tumors (Fig. 1e, f). Remarkably, *Eif3h*^-/wt mice had significantly fewer tumors, especially those with large size (≥3 mm), when compared with their *Eif3h*^flox/wt littermates (Fig. 1g). Consistently, the Ki67-positive cells in the colorectal tissues from *Eif3h*^-/wt mice were significantly decreased when compared with *Eif3h*^flox/wt littermates (Fig. 1h). Taken together, these data suggest that EIF3H deletion attenuates colitis-induced colorectal tumorigenesis.

### EIF3H promotes CRC cell growth, migration and invasion
To determine the oncogenic role of EIF3H in human CRC, we collected a series of CRC cell lines and normal human colonic epithelial NCM460 cells (Supplementary Fig. 3a). Western blot analysis showed EIF3H protein level was upregulated in CRC cells. Then we performed shRNA-mediated and doxycycline induced EIF3H knockdown (KD) in CRC cells DLD1, HCT116 and RKO, and the knockdown efficiency was confirmed by western blot and qRT-PCR (Supplementary Fig. 3b, c). EIF3H knockdown reduced the proliferation of CRC cells, as determined by cell proliferation and foci formation assays (Supplementary Fig. 3d, e). Transwell assays showed EIF3H silencing decreased the migration and invasion of CRC cells (Supplementary Fig. 3f, g). In contrast, exogenous introduction of EIF3H significantly promoted the growth and enhanced the motility of HCT116 and SW480 cells (Supplementary Fig. 3h–l). These data suggest that EIF3H promotes CRC cell growth, migration and invasion.

### EIF3H knockdown decreases HAX1 protein level in CRC cells
To elucidate the molecular mechanism and search for EIF3H substrate(s) of EIF3H-mediated CRC progression, we performed immune-affinity purification and resolved the binding partners utilizing mass spectrometry. This effort led to a number of proteins co-immunoprecipitated with EIF3H, including HAX1 and other eIF3 subunits (EIF3A-F; Fig. 2a and Table 1). Co-immunoprecipitation (co-IP) studies indicated the exogenous and endogenous interaction between EIF3H and HAX1 (Fig. 2b). In situ proximity ligation assay (PLA) confirmed EIF3H–HAX1 proximity in HCT116 cells, and most interactions between EIF3H and HAX1 were detected in the cytoplasm though few was present in the nucleus (Fig. 2c). Taken together, these data suggest that EIF3H interacts with HAX1.

To further determine whether HAX1 was a potential substrate that mediated EIF3H-promoted CRC progression, we first checked whether EIF3H could regulate HAX1 abundance. We analyzed CPTAC colon cancer cohort and TCGA-CRC cohort, and found that the HAX1 was positively correlated with EIF3H expression in CPTAC colon cancer cohort (https://cprosite.ccr.cancer.gov/) (Supplementary Fig. 4a). We observed that knockdown EIF3H in CRC cells led to a significant decrease of HAX1 protein level (steady-state expression), while no alteration in HAX1 mRNA level was observed (Fig. 2d and Supplementary Fig. 4b, c). Further, proteasome inhibitor MG132 was able to restore HAX1 protein level in CRC cells with EIF3H knockdown (Fig. 2e), indicating that EIF3H-mediated HAX1 expression was involved in proteasome-dependent mechanism. Overexpression of EIF3H prolonged the HAX1 protein half-life in HEK293T cells (Fig. 2f). Conversely, knockdown EIF3H resulted in accelerated turnover rate of HAX1 in CRC

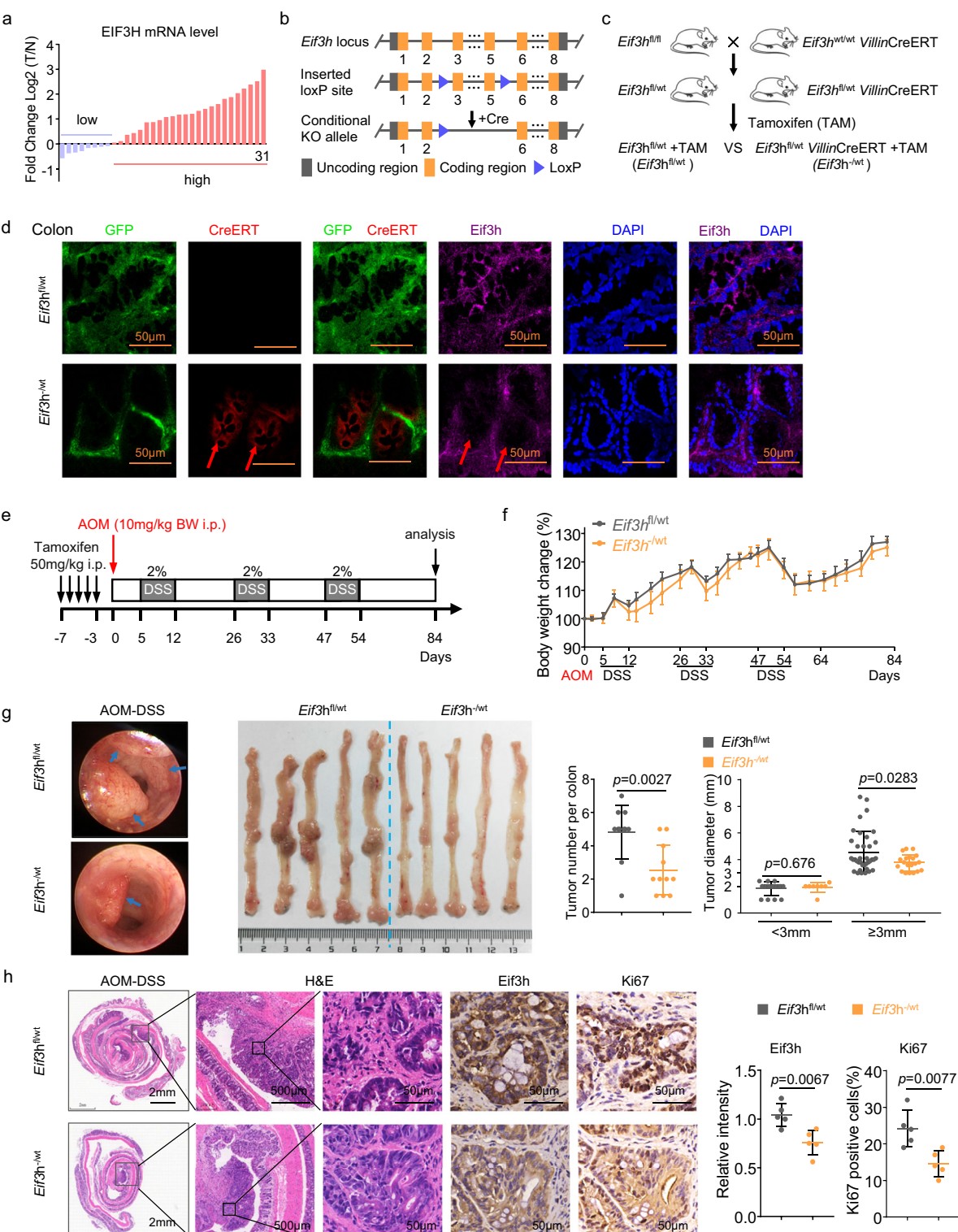

**Fig. 1 | *Villin*-specific conditional *Eif3h* knockout attenuates colorectal tumorigenesis in AOM/DSS model. a** Waterfall plot of the relative EIF3H mRNA levels from 31 paired samples of CRC and adjacent non-tumorous colorectal tissues measured by qRT-PCR. **b** Schematic depiction of generating *Eif3h* conditional *Eif3h* knockout (KO) mouse model. Exons were indicated by filled orange blocks with numbers. Blue triangle, loxP site. Filled gray blocks, uncoding region. **c** Cartoon illustration of a cross between *Eif3h*^flox/flox^ and *Villin*-CreERT to breed *Eif3h*^flox/wt^, *Villin*-CreERT mice. Tamoxifen was used to induce heterozygous conditional *Eif3h* KO. **d** Representative images of immunofluorescence staining (IF) of colon tissues obtained from the indicated tamoxifen-induced mice (*n* = 3). Nuclei stained with DAPI (blue). Scale bar = 50 μm. **e** A schematic overview of the AOM/DSS model. **f** The body weights of *Eif3h*^flox/wt^ (*n* = 7) and *Eif3h*^-/wt^ (*n* = 5) mice were measured

during the procedures. Data were presented as means ± SEM. **g** Representative mini-endoscopy pictures (left), photographs (middle), tumor numbers and tumor size of tumor bearing colons from *Eif3h*^flox/wt^ (*n* = 11) and *Eif3h*^-/wt^ (*n* = 11) mice treated with AOM/DSS. Tumors were marked by a blue arrow (left). Data were presented as means ± SD. The *p* values were calculated by two-tailed *t* test. **h** Representative hematoxylin and eosin (H&E) and immunohistochemical (IHC) staining of colon tissues obtained from the indicated mice (*n* = 5 for each group) with AOM/DSS treatment. Quantification of IHC staining were shown as bar graph (right). Data were presented as means ± SD. The *p* values were calculated by two-tailed *t* test. Source data are provided as a Source Data file. wt, wild-type. AOM, azoxymethane. DSS, dextran sodium sulphate.

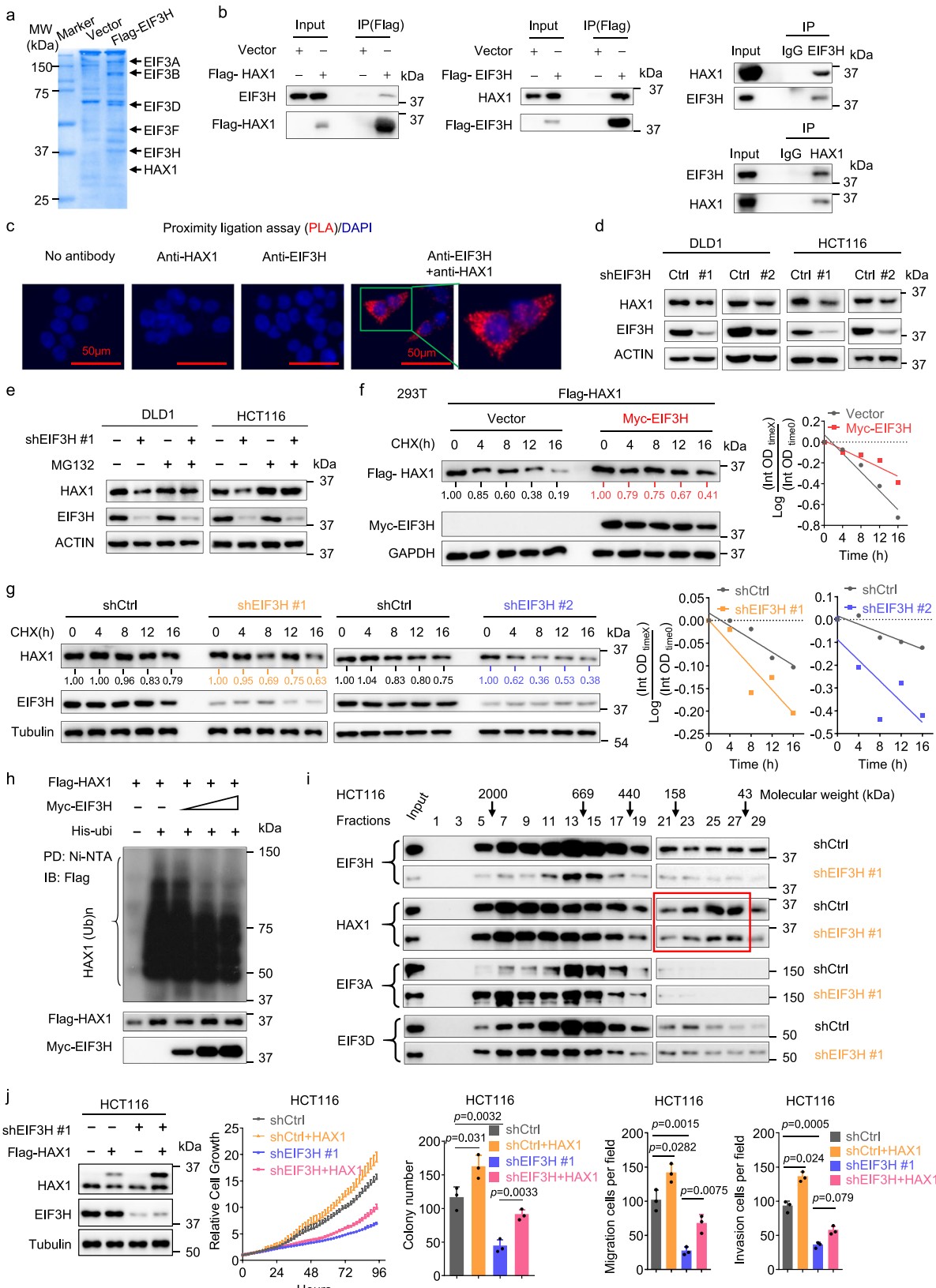

cells (Fig. 2g and Supplementary Fig. 4d). Furthermore, EIF3H over-expression decreased the ubiquitination level of HAX1 (Fig. 2h).

Previous works have reported that EIF3H controls a series of mRNA translation, such as CCND1 and ENO2[8,17]. Therefore, we further explore whether HAX1 is also a specific translational target for EIF3H. We measured the nascent protein synthesis with/without

induction of EIF3H KD in HCT116 cells and found that EIF3H KD resulted in the inhibition of nascent protein synthesis compared with the control group based on Protein Synthesis Assay Kit (Supplementary Fig. 4e, f). We also performed polysome profiling to determine the impact of EIF3H depletion on the translational efficiencies of HAX1. Interestingly, the results showed that

**Fig. 2 | EIF3H knockdown decreases steady-state expression of HAX1 protein in CRC cells. a** Coomassie blue staining of the immunoprecipitated profile using anti-Flag M2 beads in HCT116 cells transfected with empty vector or Flag-tagged EIF3H. The purified EIF3H protein complex was subjected to mass spectrometry analysis. **b** The interaction between exogenous and endogenous EIF3H and HAX1 was determined by co-IP assay. HEK293T cells were transfected with Flag-EIF3H or Flag-HAX1 plasmid. The cell lysates were pulled down with anti-Flag M2 beads and immunoblotted with the indicated antibodies. HCT116 cell lysates were immuno-precipitated with either control rabbit IgG, EIF3H, or HAX1 antibodies followed by immunoblotting. **c** Proximity ligation assay. A representative image series was shown. Red spots mark positive PLA signals for EIF3H-HAX1 interactions. Nuclei were stained with DAPI. Scale bar = 50 μm. **d** Immunoblot analysis of HAX1 expression in shCtrl and shEIF3H DLD1 and HCT116 cells. **e** shCtrl and shEIF3H CRC cells were treated with or without 25 μM MG132 for 6 h. Cell lysates were immuno-blotted. **f** HEK293T cells transfected with the indicated plasmids were treated with cycloheximide (CHX) (100 μg/mL) for indicated time. Cell lysates were immunoblotted. HAX1 levels were quantified, normalized, and the turnover of HAX1 is indicated graphically in the right panel. **g** shCtrl and shEIF3H DLD1 and HCT116 cells were treated with CHX. HAX1 turnover rate was analyzed by immunoblotting (left) and quantified (right). **h** HEK293T cells were transfected with the indicated plasmids and treated with MG132. The cell lysates were pulled down with Ni-NTA beads and immunoblotted. **i** Cellular extracts from shCtrl and shEIF3H HCT116 cells were concentrated and fractionated on Superose 6 size exclusion columns. An equal volume from each chromatographic fraction was analyzed by western-blot. Chromatographic eluate profiles and molecular size of eluted fraction were indicated. **j** shEIF3H HCT116 cells were infected with lentivirus containing control vector or Flag-HAX1 vector. The indicated protein levels, cell proliferation, colony formation, migration, and invasion were indicated. The data are presented as the means ± SD. The *p* values were obtained by two-tailed unpaired *t* test. All data shown (including immunoblotting, immunofluorescent staining) were obtained from at least three biological independent experiments with similar results. Source data are provided as a Source Data file.

EIF3H KD had no effect on translational efficiency of HAX1 mRNA, although it reduced the polysome size of CCND1, as evidenced by a shift of the polysome peak to smaller polysomes (Supplementary Fig. 4g, h).

To further support that EIF3H and HAX1 can form a complex in vivo, cellular proteins extracted from HCT116 cells were fractionated by gel-filtration chromatography. We found that EIF3H was detected in eluted fractions that largely overlapped with the HAX1 protein in fractions (#5–#29) (Fig. 2i). We further observed, while EIF3H/HAX1 cofractionated with EIF3A and EIF3D in fractions (#5–#17), EIF3H KD cell extracts had less HAX1 expression than control cell extracts when compared with fractions (#21–#29) (Fig. 2i). Meanwhile, over-expression of HAX1 rescued the EIF3H knockdown-mediated pheno-types, including cell growth, migration and invasion (Fig. 2j and Supplementary Fig. 4i). Thus, our results show that HAX1 mediates EIF3H oncogenic phenotypes in CRC cells.

## Loss-of-function EIF3H mutations attenuate its effect on stabilizing HAX1

EIF3H is classified to the JAMM deubiquitinases family for its MPN domain. We wondered whether MPN domain was essential for the recognition of HAX1 by EIF3H. We generated four truncated mutants of EIF3H as illustrated in Supplementary Fig. 5a and transfected these mutants into HEK293T cells followed by co-IP. Co-IP result showed only the full-length EIF3H and the truncated mutant EIF3H with MPN domain maintained their interaction with HAX1, while the truncated mutants without MPN domain lost the ability to interact with HAX1 (Supplementary Fig. 5a). We also mutated three residues Asp90, Asp91 and Gln121 to alanine (DDQ-AAA), located in catalytic core of the EIF3H MPN domain (Supplementary Fig. 5a), and found that EIF3H$^{DDQ/AAA}$ attenuated its interaction with HAX1 when compared with wild-type (WT) EIF3H (Supplementary Fig. 5b). However, YW-AA (Trp119, Tyr140 to alanine) mutant of EIF3H, two residues known for binding between EIF3H and YAP, still bound to HAX1 (Supplementary Fig. 5b).

We also re-introduced shRNA-resistant WT and mutant EIF3H (EIF3H$^{DDQ/AAA}$ and EIF3H$^{YW/AA}$) expression plasmids into EIF3H stable knockdown cells, and found that knockdown EIF3H could decrease steady-state expression of HAX1, while re-introduced WT EIF3H or EIF3H$^{YW/AA}$, but not EIF3H$^{DDQ/AAA}$, could rescue steady-state expression of HAX1 caused by EIF3H knockdown (Supplementary Fig. 5c). As expected, WT EIF3H and EIF3H$^{YW/AA}$ could both increase HAX1 protein half-life, while EIF3H$^{DDQ/AAA}$ failed to do so when compared with control (Supplementary Fig. 5d). More importantly, EIF3H$^{DDQ/AAA}$ mutant failed to fully rescue the proliferation, migration and invasion activities when EIF3H was knocked down in DLD1 and HCT116 cells, whereas WT EIF3H and EIF3H$^{YW/AA}$ mutant could rescue these phenomena (Supplementary Fig. 5e–g). Taken together, these results indicate that mutated EIF3H catalytic core compromises EIF3H-HAX1 interaction/regulation, and subsequent biological activities.

## E3 ubiquitin ligase βTrCP promotes HAX1 ubiquitination through binding phosphodegron of HAX1

To explore whether a specific E3 ligase is involved in HAX1 ubiquiti-nation, we used the affinity purification–mass spectrometry (MS) approach and found 4 unique peptides corresponding to βTrCP as robust interacting partners of HAX1 (Table 2). Co-IP assay confirmed the exogenous and endogenous interaction between HAX1 and βTrCP (Fig. 3a, b). βTrCP overexpression decreased the steady-state expres-sion of HAX1, and MG132 treatment could restore the expression of HAX1 protein in CRC cells with βTrCP overexpression (Fig. 3c, d). Furthermore, the half-life of HAX1 was increased in cells transfected with βTrCP specific siRNA compared with the control cells (Supple-mentary Fig. 6). Importantly, βTrCP overexpression led to accelerated turnover rate of HAX1 and increased the ubiquitination of HAX1, while the delta F-box mutant βTrCP (ΔF-box βTrCP) could not decreased the expression and stability of HAX1 (Fig. 3e, g). Intriguingly, we analyzed the HAX1 amino acid sequence and found that one putative βTrCP binding degron motif (231 DSEGRT 236) is present in HAX1 (Fig. 3h and Supplementary table 1). We generated HAX1 S232A/T236A mutant (2AA) and S232D/T236D mutant (2DD) within putative βTrCP degron[18,19]. The co-IP result indicated that HAX1$^{2AA}$ lost its binding affinity for βTrCP while HAX1$^{2DD}$ mutant enhanced its binding to βTrCP (Fig. 3h). Consistently, βTrCP acceler-ated the turnover rate of HAX1$^{2DD}$ and induced more HAX1$^{2DD}$ ubiqui-tination when compared with HAX1 WT, but had minor impact on the HAX1$^{2AA}$ mutant since it was already very stable (Fig. 3i, j). Impor-tantly, WT EIF3H could remove HAX1 polyubiquitin-conjugates mediated by βTrCP, while catalytic core mutated EIF3H (EIF3H$^{DDQ/AAA}$) failed to do so (Fig. 3k). Together, these results suggest that βTrCP increases HAX1 ubiquitination through βTrCP binding degron motif, thereby reducing the steady-state expression of HAX1. Also, βTrCP-mediated HAX1 ubiquitination can be antagonized by EIF3H with intact MPN domain.

**Table 1 | Mass spectrometry report of some EIF3H-interacting proteins in HCT116 cells**

| Accession | Description | Coverage [%] | # Peptides | # PSMs |
|-----------|-------------|--------------|------------|--------|
| O15372 | EIF3H | 60.51 | 22 | 254 |
| J9RO21 | EIF3A | 34.88 | 40 | 70 |
| B4DV79 | EIF3B | 40.79 | 21 | 40 |
| O00165 | HAX1 | 25.54 | 4 | 4 |

*PSMs* peptide-spectrum matches.

**Table 2 | Mass spectrometry detection of some HAX1-interacting proteins**

| Accession | Description | Coverage [%] | # Peptides | # PSMs | Abundances: HAX1 |
|-----------|-------------|--------------|------------|--------|------------------|
| O00165 | HAX1 | 87 | 23 | 589 | $1.005 \times 10^{11}$ |
| P04049 | RAF1 | 31 | 14 | 15 | $8.310 \times 10^{7}$ |
| Q9Y297 | βTrCP | 9 | 4 | 4 | $1.066 \times 10^{7}$ |

*PSMs* peptide-spectrum matches.

## HAX1 enhances the interaction between RAF1, MEK1 and ERK1, thereby potentiating phosphorylation/activation of ERK1/2

MS analyses also revealed that RAF1 was a potential candidate interacted with HAX1 (Table 2). Co-IP assay confirmed the exogenous and endogenous interaction between HAX1 and RAF1 (Fig. 4a). We then hypothesized that HAX1 could affect RAF1/MEK/ERK signaling in CRC. Western blot analysis showed that phosphorylation of ERK1/2 was decreased in HAX1 KD CRC cells, while phosphorylation of ERK1/2 was increased when HAX1 was overexpressed. However, there was no change in the phosphorylation level of RAF1 and MEK1/2 (Fig. 4b). The total protein levels of RAF1, MEK1/2 and ERK1/2 were unchanged, indicating that the effect of HAX1 KD is post-translational. Congruently, phosphorylation of ERK1/2 was decreased in EIF3H KD CRC cells (Fig. 4c), suggesting a link of EIF3H-HAX1-RAF1-ERK1/2.

It is possible that HAX1 exerts functions as a scaffolding protein[20]. To map RAF1-HAX1 interaction, we generated full-length and multiple truncated mutants of RAF1, as illustrated in Fig. 4d, and transfected these mutants into HEK293T cells for co-IP experiments. The mapping result showed RAF1 (301–400) amino acids residues were the domain that interacts with HAX1. To test whether HAX1-RAF1 interaction facilitates the association between RAF1, MEK1 and ERK, we co-transfected Flag-tagged RAF1 together with increasing HA-tagged HAX1. Increasing HAX1 overexpression enhanced the interaction of RAF1 with MEK1 and with ERK1/2 (Fig. 4e and Supplementary Fig. 7a). Consistently, HAX1 KD decreased the interaction between RAF1 and MEK1 in HCT116 cells (Fig. 4f). To test whether HAX1 promotes RAF1 homodimerization or heterodimerization with BRAF since RAF dimerization is critical for its activation as well as substrate phosphorylation, we performed co-IP (pull-down) experiments using different tagged proteins[21]. Cells were co-transfected with Myc-RAF1/Flag-BRAF or Flag-RAF1/Myc-RAF1 in the presence of increasing HA-tagged HAX1, then co-IP experiments were performed. Increasing amounts of HA-tagged HAX1 promoted the binding of Myc-RAF1/Flag-BRAF or Flag-RAF1/myc-RAF1 (Fig. 4g, h). Similarly, when HAX1 was knocked down, the RAF1 homodimerization and heterodimerization with BRAF was decreased, and these phenomena could be rescued by HAX1 re-expression (Fig. 4i and Supplementary Fig. 7b). Thus, our results demonstrated that increasing HAX1 enhanced the RAF1 homodimerization and RAF1/BRAF heterodimerization. Given the HAX1-EIF3H link, EIF3H KD also decreased the interaction of RAF1 with MEK1 and with ERK1/2 (Fig. 4j). Importantly, HAX1 overexpression could significantly reverse such a phenomenon caused by EIF3H KD (Fig. 4j). As for the phosphorylation activation, RAF1 overexpression restored the decrease in ERK1/2 phosphorylation caused by HAX1 KD in DLD1 and HCT116 cells (Supplementary Fig. 7d). Trametinib, a selective inhibitor of MEK1/2, has been approved by the FDA for the treatment of advanced melanoma[22]. We found that overexpression of HAX1 up-regulated the phosphorylation of ERK1/2, while MEK1/2 inhibitor trametinib attenuated the up-regulation of ERK1/2 phosphorylation caused by HAX1 overexpression (Fig. 4k). These observations provide an important insight into RAF1 signaling cascade: HAX1 is involved in the RAF1/MEK/ERK interaction. HAX1 recruits RAF1 and MEK1/2, thereby increasing the binding between MEK1/2 and ERK1/2 for facilitating phosphorylation of ERK1/2.

## EIF3H promotes CRC tumorigenesis/metastasis via HAX1 in an orthotopic CRC model

To further investigate the functional contribution of dysregulated EIF3H/HAX1 axis to CRC tumorigenesis and metastasis, we established orthotopic CRC models by injecting Tet-on-inducible-shEIF3H HCT116 cells with or without ectopic expression of HAX1 into cecum to observe the tumor growth and metastasis (Fig. 5a). We showed that EIF3H KD led to reduced tumor growth and compromised orthotopic carcinogenic as liver and lung metastatic abilities were greatly decreased when compared with control. Importantly, HAX1 overexpression increased tumorigenicity and demonstrated strong metastatic capability; therefore, it could reverse reduced tumor growth and restore compromised lung and liver metastasis caused by EIF3H KD (Fig. 5a–c). Immunohistochemistry (IHC) staining on orthotopic tumor showed that EIF3H KD led to reduced levels of EIF3H, HAX1 and pERK1/2, and repressed cell proliferation (Ki67 staining), while HAX1 overexpression reversed these impacts caused by EIF3H KD (Fig. 5d). Importantly, we found that human (colon to liver) metastatic tumors exhibited higher protein levels of EIF3H and HAX1 when compared with the paired primary tumor tissue and adjacent normal tissue from human patients with CRC (Fig. 5e). Taken together, these results suggest that EIF3H/HAX1 axis promotes CRC tumorigenesis and metastasis.

## Activation of Wnt/β-catenin signaling induces EIF3H expression in CRC cells

Wnt to β-catenin is an important signaling pathway supporting CRC development[23,24]. We analyzed change in mRNA level of *Eif3h* from datasets of colon tumors isolated from *Apc*^Min/+ and AOM-treated mice (GSE5204), and found that *Eif3h* mRNA level was elevated in tumors of *Apc*^Min/+ and AOM-induced mice compared with normal mice colon tissues (Supplementary Fig. 8a). In TCGA database, EIF3H mRNA level was positively correlated with the mRNA levels of AXIN2 and MYC, which are downstream genes of Wnt pathway (Supplementary Fig. 8b). We then speculated that activation of Wnt pathway may regulate the abnormal expression of EIF3H in CRC. Treatment of CRC cells with conditioned medium containing L-Wnt3a increased the mRNA level of EIF3H (Fig. 6a). Treatment with three inhibitors of Wnt/β-catenin signaling, ICG-001, NCB0846 and LGK974, reduced the expression of EIF3H at both mRNA and protein levels in a dose-dependent manner (Fig. 6b, c and Supplementary Fig. 8c)[25]. Further, L-Wnt3a-stimulated mRNA and protein elevation of EIF3H could also be attenuated by NCB0846 treatment (Fig. 6d, e). JASPAR website revealed several putative TCF4 binding sites within the *EIF3H* promoter (Fig. 6f). We performed chromatin immunoprecipitation (ChIP) analysis by using antibodies targeting β-catenin or TCF4 and identified 2 functional binding sites from −1752 to −1587 bp and −1155 to −990 bp of the *EIF3H* promoter (Fig. 6g). Congruently, overexpression of β-catenin and TCF4 led to the increase of EIF3H mRNA and protein expression (Fig. 6h, i). Altogether, our findings reveal that EIF3H is a transcriptional target of the β-catenin–TCF4 complex, and activation of Wnt/β-catenin signaling induced EIF3H expression. Given that activation of Wnt/β-catenin signaling induced EIF3H expression and that EIF3H-HAX1 enhanced phosphorylation of ERK1/2 in CRC, we next investigated

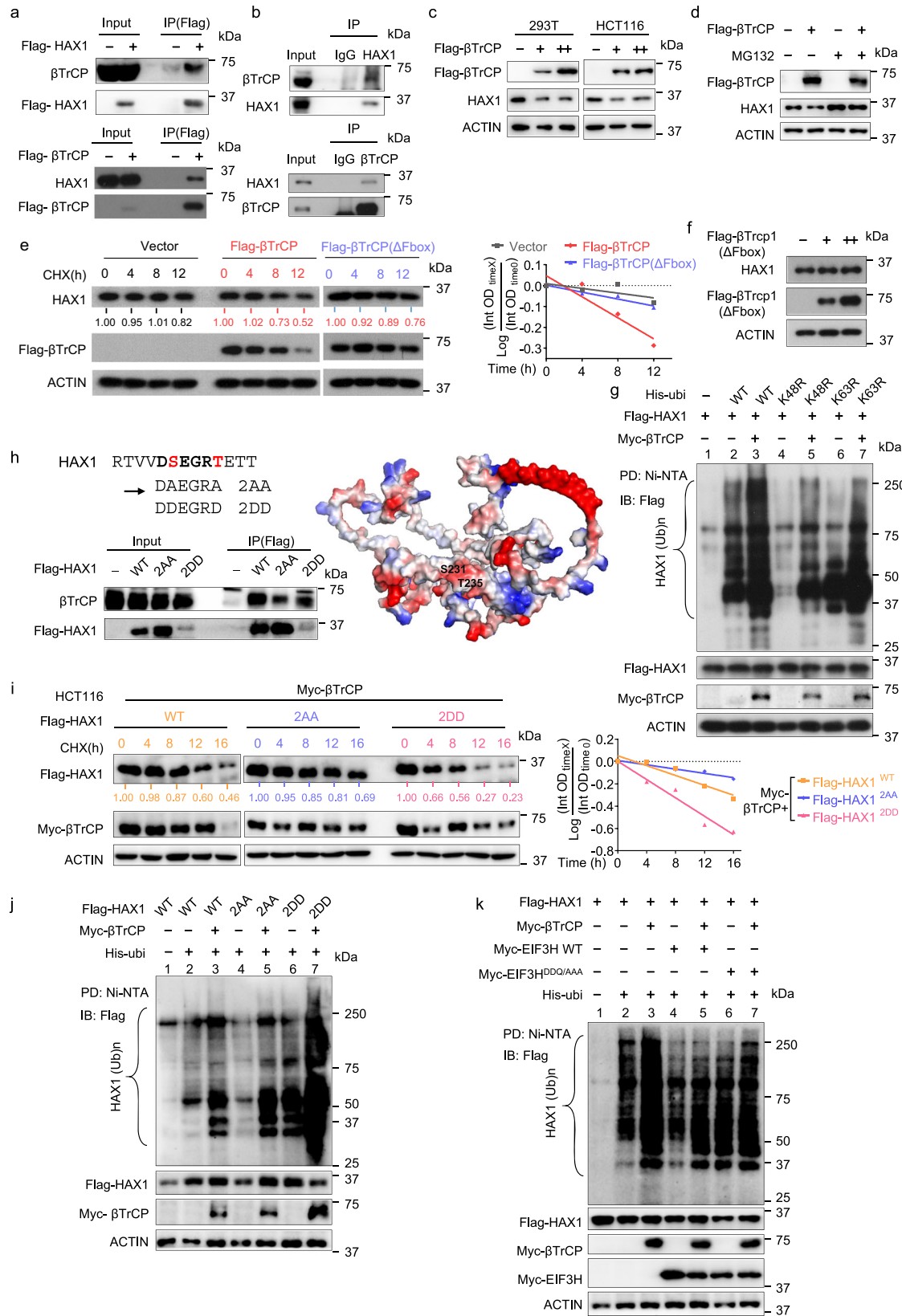

whether combined targeting Wnt and RAF1-ERK1/2 signaling synergistically inhibited CRC cell growth. We found that combined Wnt inhibitor (LGK974) and MEK1/2 inhibitor (trametinib) synergistically suppressed CRC cells colony formation in vitro in a dose-dependent manner (Fig. 6j).

**Combined Wnt and RAF1-ERK1/2 signaling inhibitors treatment suppresses tumor growth in EIF3H-high PDX with better efficacy**
IHC staining of a panel of 104 CRC and normal tissue specimens showed that EIF3H protein expression was significantly higher in CRC tissues than that in matched adjacent normal tissue (Fig. 7a). Notably,

**Fig. 3 | E3 ubiquitin ligase βTrCP promotes HAX1 ubiquitination through binding phosphodegron of HAX1. a** 293 T cells were transfected with Flag-tagged HAX1 or Flag-tagged βTrCP. Cell lysates were pulled down with anti-Flag M2 beads and immunoblotted. **b** HCT116 cell lysates were pulled down with an anti-HAX1 or anti-βTrCP antibody and immunoblotted with βTrCP or HAX1. **c** Representative immunoblots showing HAX1 steady-state expression in 293 T and HCT116 cells upon βTrCP overexpression. **d** HCT116 cells transfected with either Flag-βTrCP or vector were treated with or without MG132. Lysates were immunoblotted with indicated antibodies. **e** HCT116 cells transfected with either WT Flag-βTrCP or Flag-△F-box-βTrCP were treated with CHX for the indicated times. The cell lysates were immunoblotted (left). The turnover rate of HAX1 was shown (right).
**f** Representative immunoblots showing HAX1 steady-state expression in HCT116 cells upon △F-box-βTrCP overexpression. **g** 293T cells were cotransfected with Flag-HAX1 with or without Myc-βTrCP plus His-ubiquitin wild type (WT), K48R mutant or K63R mutant. Cells were treated with MG132 for 6 h before harvesting. The ubiquitinated HAX1 proteins were pulled down by Ni-NTA-agarose beads and detected with anti-Flag antibody. **h** Sequence alignment of the putative βTrCP-recognized degron on HAX1. 293 T cells were transfected with WT or mutant Flag-tagged HAX1 (2AA, 2DD). Location of S231 and T235 of HAX1 is indicated. The cell lysates were pulled down with anti-Flag M2 beads and immunoblotted.
**i** Representative immunoblots showing the turnover rate of HA-tagged WT or dot mutant HAX1, with or without Myc-βTrCP overexpression in HCT116 cells (left). Quantification of HAX1 turnover rate by the Image J software (right). **j, k** 293 T cells were cotransfected with the indicated plasmids. Cells were treated with MG132 for 6 h before harvesting. The ubiquitinated HAX1 proteins were pulled down by Ni-NTA beads and detected with anti-Flag antibody. Representative immunoblots shown in figures were repeated three times independently with similar results. Source data are provided as a Source Data file.

patients with high EIF3H expression had significantly reduced overall survival when compared with those with low EIF3H expression (Fig. 7a). The clinicopathological features of CRC patients are shown in Supplementary Data 1. Further analysis revealed that EIF3H expression was positively correlated with lymph nodes metastasis and was an independent prognostic factor for overall survival of CRC patients (Supplementary table 2 and 3). We established PDXs by implanting fresh CRC samples from patients into NCG mice, and chose 4 PDXs, in which two PDXs containing high EIF3H level and another two PDXs containing low EIF3H level (Fig. 7b). Next, we combined two drugs LGK974 and trametinib to treat CRC PDXs in NCG mice to assess the therapeutic potential of targeting Wnt-EIF3H pathways (Fig. 7c). As shown in Fig. 7d, the combination of LGK974 and trametinib triggered significantly growth arrest when compared with vehicle and was more effective than single drug treatment in EIF3H high-expressing CRC PDX tumors, while the efficacy of these drugs on EIF3H low-expressing PDX tumors was compromised (Fig. 7d). In addition, the administration of the combination of LGK974 and trametinib in EIF3H-high group significantly decreased Ki67-positive tumor cells and protein levels of p-ERK and c-Myc, while the impact on EIF3H-low group was less effective (Fig. 7e, f). Our data indicate that EIF3H is an actionable biomarker in CRC and that LGK974 and trametinib combination may be considered for therapeutic strategy for EIF3H high CRC patients.

## Discussion

Ubiquitin proteasome system dysregulation and the consequent aberrant activation or deactivation of signaling molecules are common hallmarks of several cancer subtypes, and targeting deubiquitinases can be a therapeutic intervention strategy for cancer[26,27]. Here we have identified that EIF3H directly deubiquitinates and stabilizes HAX1. HAX1 stabilization could enhance the binding between RAF1, MEK and ERK1/2, thereby potentiating phosphorylation of ERK1/2, which in turn promotes CRC tumorigenesis and metastasis. Activation of Wnt/β-catenin signaling induces EIF3H expression. Our results reveal upstream regulation layers of HAX1 and illustrates how oncogenic signals from Wnt-EIF3H link to βTrCP-HAX1 axis for promoting ERK activation and tumorigenesis in cancer. Combinatorial targeting of Wnt signaling and ERK1/2 shows promising effects to suppress EIF3H-high CRC growth (Fig. 7h).

Role of EIF3H in CRC remains to be further elucidated. We demonstrate that EIF3H is overexpressed in CRC tissues. Those CRC patients with high EIF3H levels have worse prognosis than those with low EIF3H levels. Our loss-of-function and gain-of-function experiments reveal that EIF3H knockdown attenuates, while EIF3H over-expression accelerates, CRC cell proliferation, migration and invasion.

To evaluate the functional role of EIF3H in CRC in vivo, we generated conditional *villin*-specific *Eif3h*-deficient mice. However, we could only generate inducible heterozygous *Eif3h* intestinal epithelial cell knockout (*Eif3h*^flox/wt^, *Villin*-CreERT) mice. Breeding did not result in progeny with inducible homozygous deletion of *Eif3h* (*Eif3h*^flox/flox^),

suggestive of embryonic lethality of *villin*-specific *Eif3h*^−/−^ mice (Supplementary Fig. 2a). This is reminiscent of the observation of another MPN containing protein CSN6[28,29]. Knockout of both *Csn6* alleles resulted in early embryonic lethality, confirming its functional importance in vivo[30]. The observation that *villin*-specific *Eif3h* homozygous knockout mice are lethal, suggesting the role of EIF3h in positively regulating the HAX1 activity, including proliferation/migration, is critical for embryo development. Nonetheless, inducible heterozygous *Eif3h* knockout (*Eif3h*^flox/wt^, *Villin*-CreERT) mice develop less and smaller colon tumors compared with their wildtype littermates in the AOM/DSS model, suggesting the effect of *Eif3h* haploinsufficiency on tumorigenesis.

EIF3H contains the JAMM motif, suggesting its deubiquitinase activity. We identify that HAX1 could interact with EIF3H. Indeed, EIF3H acts as a deubiquitinase of HAX1. HAX1 is known to be associated with the progression of various tumors, but its role in CRC is not well-characterized. We find that the phosphorylation of Ser232 and Thr236 within the HAX1 degron motif allowed βTrCP to efficiently recognize and ubiquitinate HAX1. Our studies identify βTrCP as an E3 ubiquitin ligase for HAX1, and the polyubiquitination of HAX1 mediated by βTrCP could be attenuated by EIF3H. Interestingly, it has been shown that FBXO25, another F-box protein, regulates HAX1 ubiquitination in PRKCD-dependent manner to regulate HAX1-mediated anti-apoptosis in lymphoma[14]. Also, HTLV-1 basic leucine-zipper factor (HBZ) antagonizes FBXO25 activity by blocking the interaction of HAX1 and FBXO25[31]. Whether HAX1 ubiquitination is regulated by different E3 ligases in various contexts remains to be determined. Whether EIF3H can antagonize FBXO25 or collaborate with HBZ warrants further investigation.

The dysfunction and mutants of the RAF-MEK/ERK1/2 signaling pathway are involved in tumorigenesis and metastasis and are activated in various cancers including CRC[32,33]. Importantly, the activation of the RAF-MEK-ERK cascade is very delicate and some mechanisms remain to be characterized. When RAS-GTP is low, RAF is monomeric and inactive as a consequence of intramolecular interaction between the N-terminal regulatory domain and the C-terminal kinase domain. Our results show that HAX1 binds to RAF1 directly and EIF3H or HAX1 deletion in CRC cells can reduce the phosphorylation of ERK1/2. We show that HAX1 binds to segment (300aa-400aa) of RAF1, which is the important dimerization interface of RAF1. Dimerization is thought to control the kinase domain kinetics and cause kinase activation, thereby leading to MEK recruitment and phosphorylation, and subsequent signaling down to the ERK module. We show that HAX1 recruits RAF1 and MEK1/2, and then increases the binding between MEK1/2 and ERK1/2 for phosphorylation of ERK1/2. HAX1 depletion impairs the interaction between RAF1, MEK1/2, and ERK1/2, resulting in a decrease in the phosphorylation of ERK1/2. It is possible that HAX1 binding changes the conformational flexibility of inactive RAF1 kinases and facilitates the alignment of two parallel regulatory and catalytic spines columns with conserved hydrophobic residues, thereby

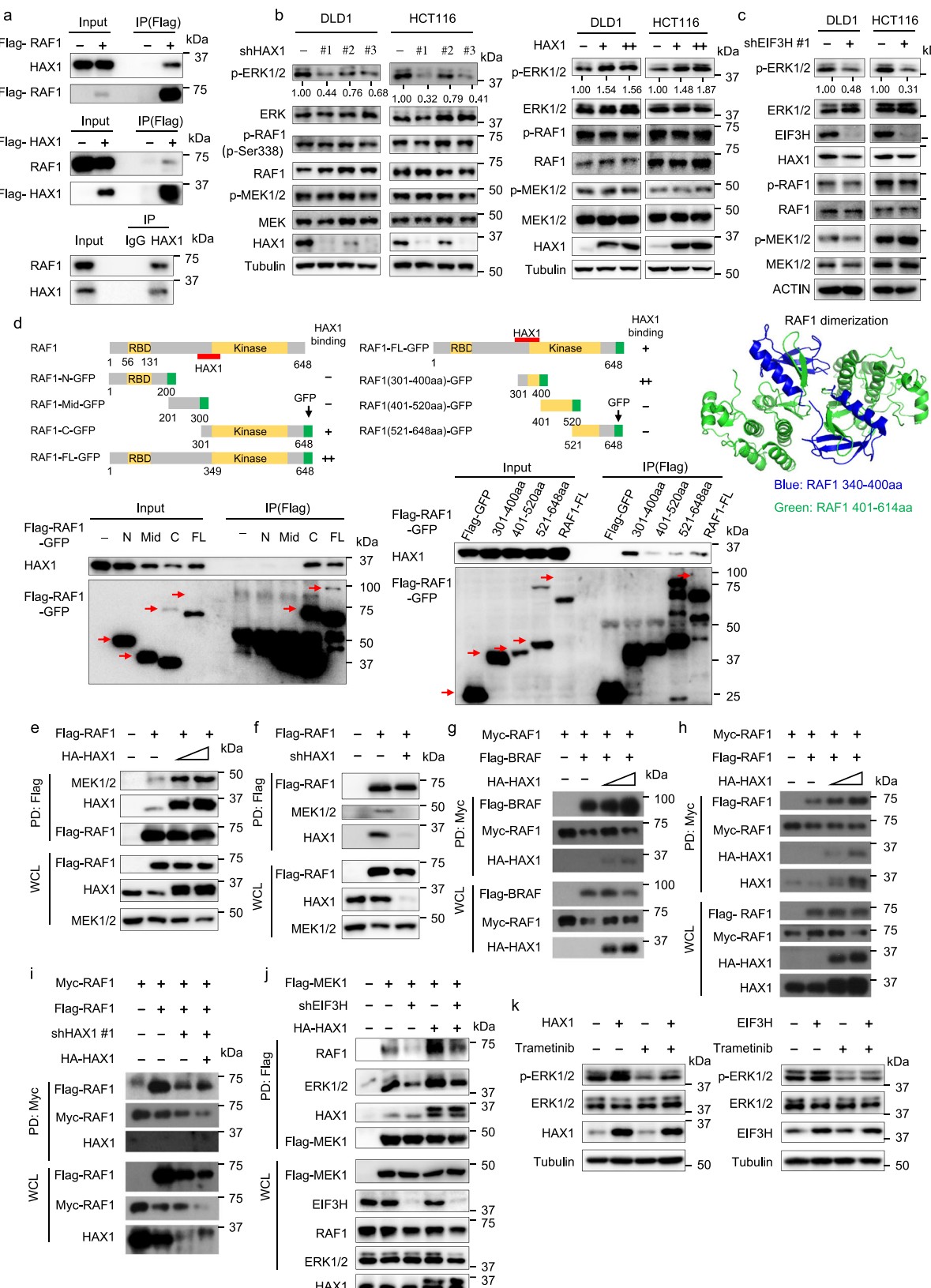

instigating catalytic activation of RAF1[33]. Obviously, the dimer interface of RAF can serve as target to develop inhibitors for RAF inactivation. Therefore, targeting HAX1 may also attenuate the activation of RAF.

In the orthotopic CRC model, EIF3H knockdown decreases orthotopic tumor growth, and attenuates liver and lung metastasis, while HAX1 can rescue these phenomena under EIF3H knockdown. It is

not clear how HAX1 can promote colon to liver/lung metastasis in mouse model. However, HAX1's impact on CRC metastasis is quite impressive not only in mouse model but also in human cancer tissue (Fig. 5e). It is conceivable that impact of HAX1 in RAS-RAF-MEK-ERK pathway can override tightly regulated mechanisms to facilitate tumor proliferation and metastasis. It has been shown that BRAF$^{V600E}$ tumors

**Fig. 4 | HAX1 enhances the interaction between RAF1, MEK1, and ERK1, thereby potentiating phosphorylation/activation of ERK1/2. a** The interaction between HAX1 and RAF1 was determined by endogenous and semi-exogenous co-IP assay. (Top and middle) HEK293T cells were transfected with Flag-HAX1 or Flag-RAF1. The cell lysates were pulled down with anti-Flag M2 beads and immunoblotted with the indicated antibodies. (Bottom) HCT116 cell lysates were pulled down with an anti-HAX1 antibody and immunoblotted with RAF1. **b** Effect of HAX1 knockdown and overexpression on the phosphorylation of ERK in DLD1 and HCT116 cells was detected by western blotting. **c** Effect of EIF3H knockdown on the phosphorylation of ERK in DLD1 and HCT116 cells was detected by western blotting. **d** Full length and truncated mutant Flag-RAF1-GFP plasmids were transfected into 293T cells. Cell lysates were immunoprecipitated with anti-Flag M2 beads and immunoblotted with indicated antibodies for binding studies. RAF1 dimerization domain in RAF1 structure was indicated (340–400aa). **e** The HEK293T cells were transfected with Flag-RAF1 with increasing HA-HAX1 plasmids. Cell lysates were immunoprecipitated with anti-Flag M2 beads and immunoblotted with indicated antibodies.

WCL, whole cell lysis. **f** The shCtrl and shHAX1 HCT116 cells were transfected with Flag-RAF1 plasmids. Cell lysates were immunoprecipitated with anti-Flag M2 beads and immunoblotted with indicated antibodies. **g, h** The HEK293T cells were transfected with Myc-RAF1, Flag-BRAF **g** or Flag-RAF1 **h** with increasing HA-HAX1 plasmids. Cell lysates were immunoprecipitated with anti-Myc beads and immunoblotted with indicated antibodies. **i** The shCtrl and shHAX1 HCT116 cells were transfected with Myc-RAF1, Flag-RAF1 with or without HA-HAX1 plasmids. Cell lysates were immunoprecipitated with anti-Myc beads and immunoblotted with indicated antibodies. **j** The shCtrl and shEIF3H HCT116 cells were transfected with Flag-RAF1 or HA-HAX1 plasmids. Cell lysates were immunoprecipitated with anti-Flag M2 beads and immunoblotted with indicated antibodies. **k** The overexpression of HAX1 or EIF3H HCT116 cells were treated with or without Trametinib. Western blot analysis of indicated proteins were performed. Representative immunoblots shown in figures were repeated three times independently with similar results. Source data are provided as a Source Data file.

have high levels of glutamate-cysteine ligase (GCL), and form distant liver and lung metastases. It remains to be characterized whether GCL gets involved in HAX1-mediated metastasis[34]. Also BRAF$^{V600E}$ oncoprotein increases the expression of miR-222-3p and Snail, thus it is possible that HAX1 may regulate metastasis via miR-222-3p and Snail axis[35].

Our findings revealed that EIF3H is a target of Wnt/β-catenin signaling and that it has links between the MEK/ERK and Wnt/β-catenin pathways through EIF3H/HAX1 axis. We then hypothesized that combination approaches targeting Wnt signaling (Wnt inhibitor, LGK974) and MEK/ERK (MEK inhibitor, trametinib) pathways may have a better synergistic efficacy in treating EIF3H-high CRC. Indeed, combined inhibition of MEK1/2 (trametinib) plus Wnt (LGK974 inhibitor) leads to the strong growth inhibitory effect than trametinib or LGK974 alone in EIF3H-high CRC PDX model. Thus, these findings provide pivotal prognostic and therapeutic implications for improving therapeutic efficacy of those CRC patients with high expression of EIF3H.

In summary, we highlight the functional importance of EIF3H/HAX1 axis in mediating CRC growth and metastasis through regulating MEK/ERK1/2 signaling. Therefore, EIF3H/HAX1 may serve as a promising biomarker and potent molecular target for CRC treatment. Further, HAX1 participates in RAF-MEK-ERK and Wnt pathway, thus targeting HAX1 is appealing for the development of anti-cancer therapeutics. Future screening of inhibitors targeting EIF3H/HAX1 or molecules blocking EIF3H-HAX1 binding may provide a promising therapeutic approach for CRC treatment.

## Methods

All animal experiments were approved by the Animal Ethical and Welfare Committee of the Sixth Affiliated Hospital of Sun Yat-sen University and carried out following its legal requirements. The collection and use of clinical samples were in accordance with research ethics board approval from the Sixth Affiliated Hospital of Sun Yat-sen University Review Board.

### Patients and specimens

Thirty-one CRC tumor tissues and matched adjacent non-tumorous colorectal tissues, in which 3 cases with liver metastasis, were collected from the Department of Colorectal Surgery at the Sixth Affiliated Hospital of Sun Yat-sen University with the patients' written informed consent and approval from the Institutional Review Board of the Sixth Affiliated Hospital of Sun Yat-sen University[36]. The human colorectal cancer tissue array (HColA180Su16) was purchased from Shanghai Outdo Biotech Co., Ltd., which contains 104 CRC cases (76 cases containing tumor and matched adjacent normal tissue, 28 cases only containing tumor tissues) and the related clinical and survival information[37].

### Transgenic mice and AOM/DSS model

Male *Eif3h*$^{flox/wt}$ mice on the C57BL/6 background were generated by the CRISPR/Cas9 method (GemPharmatech Co., Ltd., China). *Eif3h*$^{flox/wt}$ mice were intercrossed with *Villin*-CreERT mice to obtain *Eif3h*$^{flox/wt}$, *Villin*-CreERT mice, as *Eif3h*$^{flox/flox}$, *Villin*-CreERT mice were not viable. To induce heterozygous conditional *Eif3h* knockout, the female *Eif3h*$^{flox/wt}$, *Villin*-CreERT mice were intraperitoneally injected with 50 mg/kg tamoxifen at an age of 4–5 weeks for five consecutive days. Three days after the last injection of tamoxifen, *Eif3h*$^{flox/wt}$ mice and *Eif3h*$^{flox/wt}$, *Villin*-CreERT mice were injected once with AOM (10 mg/kg, intraperitoneally). 5 days later, mice were given 2% dextran sulfate sodium (DSS) (MP Biomedicals) in drinking water for 7 days followed by 2 weeks of regular drinking water. DSS treatments were repeated for two additional cycles, and mini-endoscopy was used to monitored tumor development. The mice were sacrificed on day 84 after the AOM injection.

### Mouse genotyping

Mouse tail tips were collected at 14-day-old, boiled in lysis buffer A (50 mM NaOH, 0.2 mM EDTA) at 99 °C for 30 min, and then neutralized with buffer B (40 mM Tris-HCl, pH 6.8). Then, 1 μL of extracted genomic DNA was used for further PCR with following primers: *Villin*- Forward: GTGTGGGACAGAGAACAAACC, *Villin*-Reverse: ACATCTTCAGG TTCTGCGGG; *Eif3h*−5'F: AGCCATTTCTCCAGTCTTGTATTGC, *Eif3h* −5'R: ACCAGACACAGGCTACAGCTAATTC. *Eif3h* primers yield a 328 bp band for wild type locus and a 433 bp band for floxed allele.

### Cell culture, reagents, and transfection

All the cells were obtained from ATCC, and maintained at 37 °C in 5% CO$_2$ incubators. DLD1 (CCL-221), SW480 (CCL-228), HCT116 (CCL-247), HCT8 (CCL-244), HT29 (HTB-38), Caco-2 (HTB-37), SW620 (CCL-227) and LoVo (CCL-229) cells were cultured in RPMI 1640 medium supplemented with 10% (v/v) fetal bovine serum (FBS). HEK293T (CRL-3216) were cultured in Dulbecco's modified Eagle's medium media with 10% FBS. RKO (CRL-2577) were cultured in Minimum Essential Medium with 10% FBS. Colon normal immortalized epithelial cell line NCM460 was obtained from In Cell (San Antonio, TX). For transient transfection, plasmids were transfected into cell lines followed the standard protocol for Lipofectamine 2000 Transfection Reagent (Thermo Fisher, #11668019). ICG-001 and NCB0846 were purchased from Selleck (S2662 and S8392).

### Plasmids construction

Plasmids pcDNA3.1-myc-EIF3H (1–352, full length), pcDNA3.1-Flag-EIF3H(1–352, full length), truncated mutants Flag-EIF3H were constructed by substitution of PCR-amplified each EIF3H fragment into the EcoRI sites of pcDNA3.1-myc or pcDNA3.1-Flag vector. Plasmids pcDNA3.1-Flag-HAX1, pcDNA3.1-HA-HAX1, pcDNA3.1-myc-βTrCP,

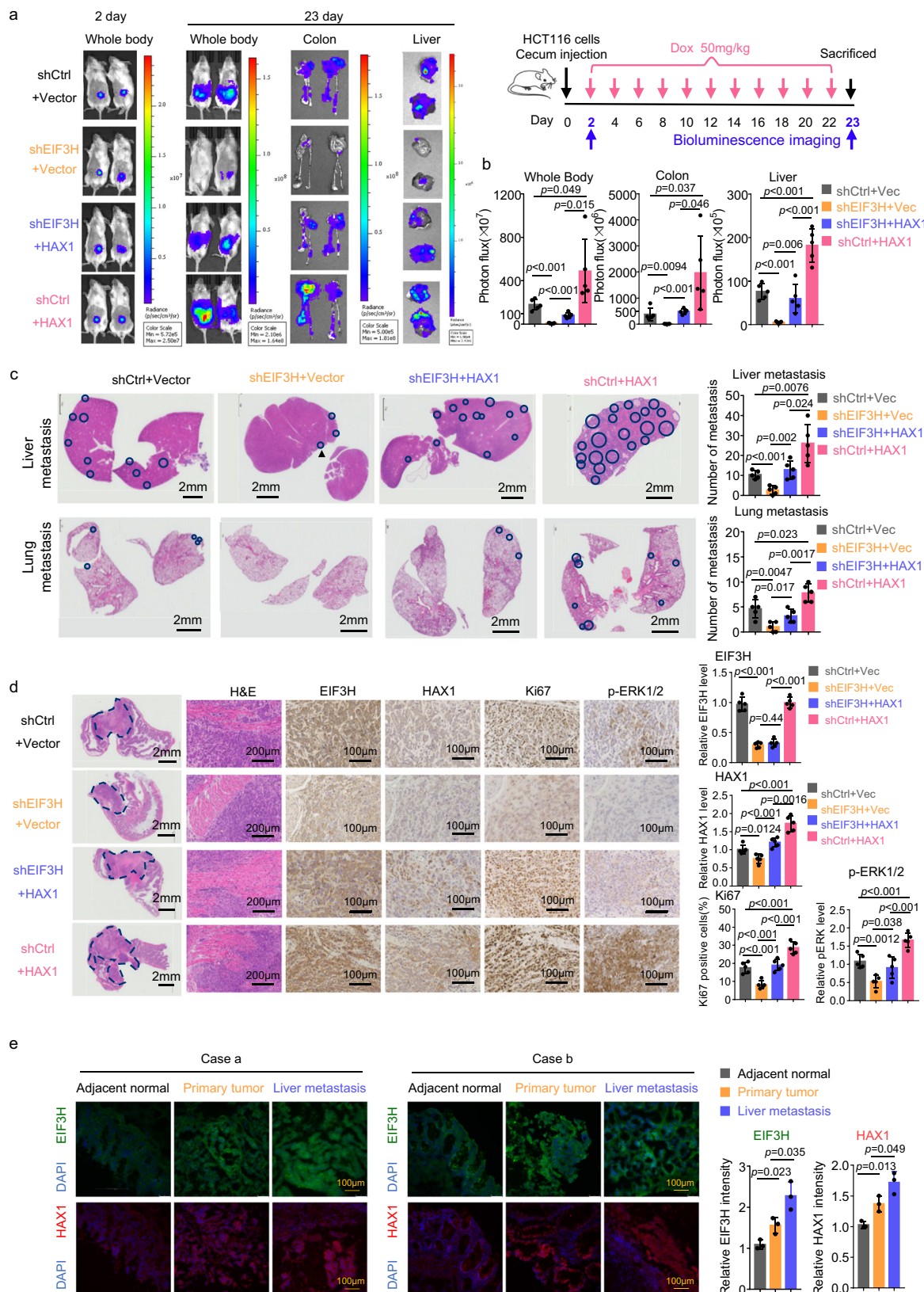

pcDNA3.1-Flag-βTrCP, pcDNA3.1-Flag-F-box mutant βTrCP1, pcDNA3.1-Flag-βTrCP2, pcDNA3.1-Flag-FBXO1, pcDNA3.1-SKP2, pcDNA3.1-HA-FBXO25, pcDNA3.1-Flag-β-catenin, pcDNA3.1-Flag-TCF4, pcDNA3.1-Myc-RAF1, pcDNA3.1-Flag-BRAF and pcDNA3.1-Flag-RAF1 were constructed by the same way. For stable EIF3H and HAX1 overexpression, Flag-EIF3H and Flag-HAX1 sequence were PCR-amplified from pcDNA3.1-Flag-EIF3H and pcDNA3.1-Flag-HAX1 and subcloned into the EcoRI sites of pLVX-Puro plasmid. EIF3H dot mutation plasmids were generated by replacement as D90A, D91A, Q121A, W119A and Y140A of pcDNA3.1-myc-EIF3H by site-directed mutagenesis using Mut Express II Fast Mutagenesis Kit (Vazyme Biotech) according to the manufacturer's instructions. HAX1 dot mutation plasmids were generated by

**Fig. 5 | EIF3H promotes CRC tumorigenesis/metastasis via HAX1 in an orthotopic CRC model. a** Schematic diagram of orthotopic xenograft CRC model (right). The HCT116 cells were orthotopically inoculated into the cecum of mice (n = 5 for each group). At day 2 and 23 after inoculation, the bioluminescent images were captured and quantified. Representative bioluminescence images of colorectal orthotopic inoculation mice, isolated intestines and livers (left). **b** Data of orthotopic CRC model (n = 5 for each group) were presented as mean ± SD. The p values were obtained by one-way ANOVA. **c** Representative images of H&E staining on liver and lung tissue sections (left) and quantification of metastatic tumor areas (right) (n = 5 for each group). The dark blue circles indicated the tumor borders; data were presented as means ± SD. The p values were calculated by one-way ANOVA.

**d** Representative images of H&E staining and IHC staining for EIF3H, HAX1, Ki67, and pERK1/2 on orthotopic CRC sections (n = 5). The dark blue dashed lines indicated the tumor borders. Quantifications of IHC staining were shown as bar graphs. Data were presented as means ± SD. The p values were calculated by one-way ANOVA. **e** Representative images of immunofluorescence staining for EIF3H and HAX1 in primary CRC tumor, adjacent normal, and liver metastasis tissue obtained from the same CRC patient (left). Green, EIF3H; red, HAX1; blue, DAPI. The staining intensity of EIF3H and HAX1 was quantitated by ImageJ and presented as bar graphs (right) (n = 3). Data were presented as means ± SD. The p values were calculated by one-way ANOVA. Source data are provided as a Source Data file.

replacement as S232A, S232D, S236A and S236D of pcDNA3.1-Flag-HAX1 by site-directed mutagenesis. Plasmids Flag-RAF1-GFP and a series of truncated mutant Flag-RAF1-GFP were constructed by substitution of PCR-amplified each RAF1 fragment into the BamHI sites of pEGFP-N plasmid. For EIF3H and HAX1 knockdown, the shRNA target sequences were inserted into pLKO.1 puro vector (Supplementary Table 4). EIF3H shRNA #1 was designed to target 3′UTR of EIF3H. For the doxycycline-inducible EIF3H KD, the shRNAs targeting EIF3H were inserted into Tet-pLKO-puro vector. All restriction enzymes were purchased from New England Biolab (MA, US).

### shRNA or siRNA knockdown

Lentiviral particles were generated by transfecting HEK293T cells with pLKO.1 shRNA or Tet-pLKO-puro vector construct and packaging vectors (psPAX2 and pMD2.G). Then the cancer cells were infected twice with culture medium containing lentivirus in the presence of 8 mg/mL polybrene. 48 h after infection, selection of transduced cells was performed with puromycin to increase the knockdown efficiency. siRNA were transfected into cell lines followed the standard protocol for Lipofectamine 2000 Transfection Reagent (Thermo Fisher, #11668019). All shRNA and siRNA sequences used in this study are listed in Supplementary Table 4.

### Western blotting and immunoprecipitation

Cells were collected from cultured dishes and lysed in ice-cold lysis buffer (20 mM Tris-HCl pH 7.5, 1 mM EDTA, 150 mM NaCl, 1% Triton-100) containing protease inhibitor cocktail and phosphatase inhibitor cocktail (bimake). Protein lysates were quantified using a BCA Protein Assay Kit (Servicebio) and equal amounts of protein were loaded on SDS-PAGE gel, transferred onto a PVDF membrane (Cat #IPVH00010, Millipore, Billerica, MA, USA) and immunoblotted with a primary antibody at 4 °C overnight. After washing with TBS-T, the membrane was incubated for 1 h with horseradish peroxidase (HRP)-conjugated secondary antibody (Goat anti-Mouse IgG (H + L), ThermoFisher 31430; Goat anti-Rabbit IgG (H + L), ThermoFisher 31460; 1:10000), and then detected by enhanced chemiluminescence (ECL) system followed by exposure to the Bio-Rad ChemiDoc imaging system. Antibodies against EIF3H (Cell Signaling, 3413, 1:4000), HAX1 (Proteintech, 11266-1-AP, 1:1000), EIF3A (Cell Signaling, 3411, 1:1000), EIF3D (Santa Cruz, sc-271515, 1:500), Phospho-ERK (Cell Signaling, 4370, 1:1000), ERK (Cell Signaling, 4695, 1:2000), c-Myc (Cell Signaling, 13987 S, 1:1000), AXIN2 (Proteintech, 20540-1-AP, 1:1000), Flag-tag (Sigma, F1804, 1:4000), HA-tag (Cell Signaling, 3724 S, 1:2000), Myc-Tag (9B11) (Cell Signaling, 2276 s, 1:4000), βTrCP (D13F10) (Cell Signaling, 4394 S, 1:1000), RAF1 (Cell Signaling, 53745, 1:1000), MEK1/2 (Cell Signaling, 8727, 1:2000), Phospho-MEK1/2 (Ser217/221) (41G9) (Cell Signaling, 9154, 1:1000), Phospho-RAF1 (Ser338) (56A6) (Cell Signaling, 9427, 1:1000), beta-Actin (sigma, A5441, 1:1000), alpha-Tubulin (Proteintech, 66031-1-Ig, 1:4000) and GAPDH (Proteintech, 60004-1-Ig, 1:10000) were used in this study.

For Co-IP experiments, cells were lysed by ice-cold NP-40 buffer containing 50 mM Tris–HCl (pH 7.5), 1 mM EDTA, 150 mM NaCl, 0.1%

Nonidet P-40, 0.1% Triton-100, protease inhibitor cocktail and phosphatase inhibitor cocktail (bimake). For endogenous Co-IP experiments, cells lysates (500 μL) were incubated with primary antibody under 4 °C overnight and Protein A/G Sepharose. Rabbit IgG (Sigma, I5006) was used as control. For Flag or MYC immunoprecipitation, cell lysates were incubated with anti-Flag M2-beads (Sigma, A2220) or Anti-Myc Magnetic Beads at 4 °C for 3-4 h with gentle shaking. The beads were washed five times with NP-40 buffer, followed by SDS-PAGE and western blot analysis.

### Mass spectrometry

HCT116 cells stably expression Flag-tagged EIF3H or HAX1 were washed twice with PBS and lysed with ice-cold NP-40 buffer. EIF3H and HAX1 interacting proteins were purified and eluted by M2 beads immunopurification and washing respectively. The elute was then separated on SDS-PAGE followed by Coomassie blue staining. The affinity purified EIF3H or HAX1 associated proteins were analyzed by liquid chromatography (LC)-MS/MS in Shanghai Applied Protein Technology Co., Ltd. LC-MS/MS analysis was performed on a Q Exactive mass spectrometer (Thermo Scientific) that was coupled to Easy nLC (Proxeon Biosystems, now Thermo Fisher Scientific). The peptides were loaded onto a reverse phase trap column (Thermo Scientific Acclaim PepMap100, 100 μm × 2 cm, nanoViper C18) connected to the C18-reversed phase analytical column (Thermo Scientific Easy Column, 10 cm long, 75 μm inner diameter, 3μm resin) in buffer A (0.1% Formic acid in water) and separated with a linear gradient of buffer B (84% acetonitrile and 0.1% Formic acid) at a flow rate of 300 nl/min. The mass spectrometer was operated in positive ion mode. MS data was acquired using a data-dependent top20 method dynamically choosing the most abundant precursor ions from the survey scan (300–1800 m/z) for HCD fragmentation. The automatic gain control (AGC) target was set to 1e6, and maximum inject time to 50 ms. Dynamic exclusion duration was 30.0 s. Survey scans were acquired at a resolution of 60,000 at m/z 200 and resolution for HCD spectra was set to 15,000 at m/z 200, and isolation width was 1.5 m/z. Normalized collision energy was 30 eV and the underfill ratio, which specifies the minimum percentage of the target value likely to be reached at maximum fill time, was defined as 0.1%. The instrument was run with peptide recognition mode enabled. The MS raw data for each sample were combined and searched using the MaxQuant 1.6.14 software for identification and quantitation analysis.

### Immunofluorescence

Frozen sections were fixed with 4% paraformaldehyde for 15 min and then blocked with 2% bovine serum albumin for 1 h at room temperature. After blocking, sections were incubated with EIF3H (Cell Signaling, 3413, 1:2000) and HAX1 (Proteintech, 11266-1-AP, 1:500) antibodies at 4 °C overnight. The samples were subsequently incubated with Alexa Fluor 488-, Alexa Fluor 594- or Alexa Fluor 647-conjugated secondary antibodies (Invitrogen, 1:800). DAPI was then used for counterstaining the nuclei. Fluorescence signals were imaged on the Zeiss LSM 710 Multiphoton Laser Scanning Confocal.

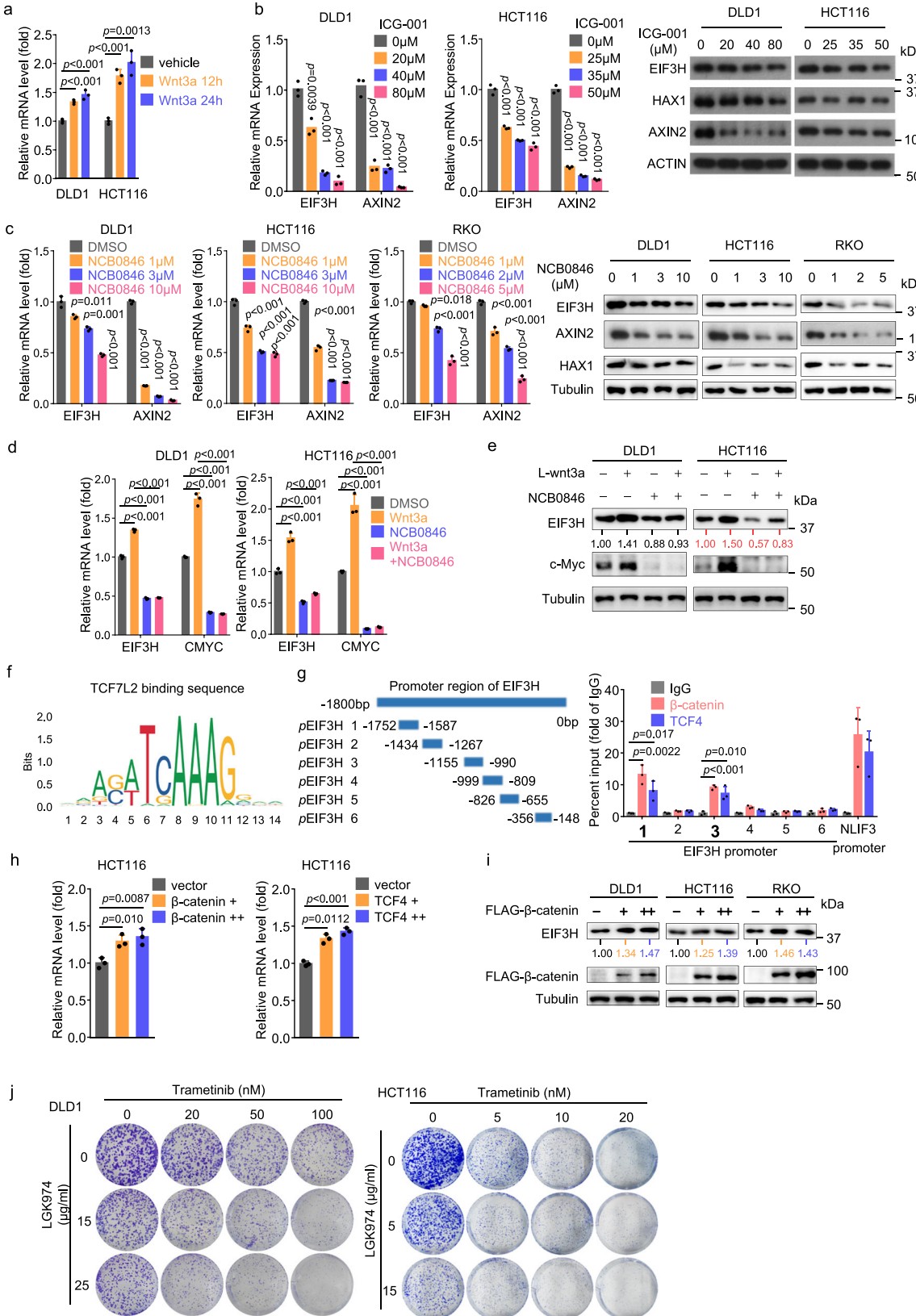

## Immunohistochemistry staining

After deparaffinization and rehydration, paraffin-embedded tissue sections (4 µm thick) were treated with retrieval buffer (pH 6.0 sodium citrate buffer) in microwave oven for 10 min. After cooling down, the sections were incubated in 3% $H_2O_2$ for 10 min to quench endogenous peroxidase activity and blocked with goat serum for 1 h. Then the sections were incubated with primary antibody overnight at 4 °C followed by incubation with secondary antibodies. Staining was visualized with diaminobenzidine (Zhongshan Goldenbridge Biotechnology). Then, sections were counterstained with hematoxylin, dehydrated, and covered with coverslip. Then the sample staining score was calculated by multiplying the percentage score and the

**Fig. 6 | Activation of Wnt/β-catenin signaling induces EIF3H expression. a** qRT-PCR analysis of EIF3H mRNA in DLD1 and HCT116 cells treated with L-Wnt3a-expressing cell conditioned medium. Data are presented as the means ± SD, $n$ = 3 biologically independent experiments. **b** qRT-PCR and western-blot analysis of EIF3H and AXIN2 levels in DLD1 and HCT16 cells treated with different concentrations of Wnt pathway inhibitor ICG-001. Data are presented as the means ± SD, $n$ = 3 biologically independent experiments. **c** qRT-PCR and western-blot analysis of EIF3H and AXIN2 levels in DLD1, HCT16, and RKO cells treated with different concentrations of Wnt pathway inhibitor NCB0846. Data are presented as the means ± SD, $n$ = 3 biologically independent experiments. **d, e** qRT-PCR **d** and western-blot **e** analysis of EIF3H and c-Myc level in DLD1 and HCT116 cells cultured with L-Wnt3a-expressing cell CM, in the presence or absence of Wnt pathway inhibitor NCB0846 (10 μM). Data are presented as the means ± SD, $n$ = 3 biologically independent experiments. **f** The potential binding sequence of TCF4 obtained from JASPAR website. **g** Deletion mutants of the EIF3H promoter (middle); HCT116 cells were subjected to chromatin immunoprecipitation using antibodies against IgG, TCF4, or β-catenin, followed by qRT-PCR for the loci on EIF3H promoter (right). NLIF3 was a positive control. Data are presented as the means ± SD, $n$ = 3 biologically independent experiments. **h, i** CRC line DLD1, HCT116 and RKO cells were transfected with Flag-TCF4 or Flag-β-catenin plasmids. qRT-PCR and western-blot analysis of EIF3H level. Data are presented as the means ± SD, $n$ = 3 biologically independent experiments. **j** DLD1 and HCT116 cells were treated with different concentrations of Wnt pathway inhibitor LGK974 plus MEK1/2 inhibitor trametinib. The colony formation assay was performed. The $p$ values were determined by unpaired two-tailed $t$ test for **a–d**, **g**, **h**. Representative immunoblots shown in the figures were repeated three times independently with similar results. Source data are provided as a Source Data file.

intensity score. Ki67 (Cell Signaling, 9449, 1:800), EIF3H (Cell Signaling, 3413, 1:2000), HAX1 (Proteintech, 11266-1-AP, 1:800), Phospho-ERK1/2 (Cell Signaling, 4370, 1:400) antibodies were used. The immunoreactive score ranged from 0 to 12 is calculated by of multiplying the positive-cell–proportion score (0–4) and the staining-intensity score (0–3).

### In vivo ubiquitination assay
HEK293T and HCT116 cells were co-transfected with a His-ubiquitin vector in combination with the indicated constructs. 48 h after transfection, cells were treated with 25 μM MG132 (Sigma) for 6 h, and then lysed in denaturing buffer A (6 M guanidine-HCl, 0.1 M $Na_2HPO_4$/$NaH_2PO_4$, 10 mM imidazole) and sonicated. The cell lysates were incubated with Nickel-nitrilotriacetic acid (Ni-NTA) agarose beads (Qiagen) overnight at 4 °C. The His pull-down products were washed twice with buffer A, twice with buffer A/TI [25 mM tris-HCl and 20 mM imidazole (pH 6.8)] (1 volume buffer A and 3 volumes buffer TI), and twice with buffer TI. The pull-down proteins were subjected to SDS-PAGE for immunoblotting.

### Polysome profiling
For polysome profiling[38], $5 × 10^7$ HCT116 cells were treated with 100ug/ml cycloheximide for 15 min and then lysed by polysome cell extraction buffer (50 mM MOPS, 15 mM $MgCl_2$, 150 mM NaCl, 100 mg/ml cycloheximide, 0.5% Triton X-100, 1 mg/ml Heparin, 200U/ml RNase inhibitor, 2 mM PMSF and 1 mM Benzamidine) for 10 min on the ice and centrifuged at 13,000 x g for 10 min at 4°C. After that, 1 ml of cytoplasmic extract was layered onto 11 ml of 10%- 50% Sucrose gradient and then centrifuged at 36,000 x rpm at 4°C for 3 h in a SW41 rotor (Beckman Coulter, USA). Separated samples were fractionated at 0.75 ml/min through BR-188 Density Gradient Fractionation System (Brandel, USA) and monitored at absorbance 254 nm. Monosome and polysome fractions were collected for RNA isolation to study the relative distribution of HAX1, CCND1 and ACTB mRNAs. The RNA in each fraction was extracted using RNA-Quick Purification Kit (ESscience).

### Evaluation of nascent protein synthesis
Appropriated number of DLD1 and HCT116 cells were plated into 96-well plates with/without doxycycline induction of EIF3H KD for 48 h. The Click-iT® Plus OPP Alexa Fluor® 488 Protein Synthesis Assay Kit (Life Technologies, Grand Allen, NY) was used for the detection of protein synthesis utilizing Incucyte S3 imaging system[39]. Cells were further applied to fixation and subsequently following the protocol provided by the manufacturer. The Alexa Fluor® 488 fluorescence intensity of all the cells were automatically quantified by high-content imaging, which corresponds to protein synthesis. As a control, cells were treated with cycloheximide for 1 h before OPP addition. The data are presented as the average protein synthesis in bar graphs, and representative images of DLD1 and HCT116 cells are depicted in the figure.

### Gel filtration chromatography
HCT116 cells were lysed in lysis buffer (50 mM Tris pH 7.5, 0.5% Nonidet P-40, 0.1% Triton X-100, 150 mM NaCl, 0.1 M EDTA) containing protease inhibitor cocktail and phosphatase inhibitor cocktail. Approximately 5 mg protein was concentrated to 0.5 mL and loaded into the Superose 6 Increase 10/300 GL (Cytiva, 29091596) and then fractionated with the cold PBS buffer (0.01 M phosphate buffer, 0.14 M NaCl, pH.7.4) in GE AKTA avant 150 chromatography system. Elution was carried out at a flow rate of 0.4 mL/min and fractions were collected every 300 μL. Collected fractions were analyzed by SDS-PAGE and Western blot with the indicated antibodies.

### In situ Proximity Ligation Assay (PLA)
The In situ Proximity Ligation Assay was performed according to the manufacturer's instructions (SigmaAldrich, DUO92101). The fixed HCT116 cells were first blocked with the blocking solution for 60 min and then incubated with primary antibodies (rabbit anti-HAX1 antibody (Proteintech, 11266-1-AP) and mouse anti-EIF3H antibody (GeneTex, GTX633631)) at 4 °C overnight. Then, cells were incubated with secondary proximity probes at 37 °C for 90 min. Ligation mix was then applied to each of the sample to complete the ligation process at 37 °C for 30 min. Samples were then incubated with the Polymerase for the amplification and incubated at 37 °C for 100 min. After final washes, the samples were mounted using Duolink® In Situ Mounting Medium with DAPI and imaged with a Zeiss LSM 710 Multiphoton Laser Scanning Confocal.

### RNA isolation and quantitative real-time PCR
Total RNA was extracted from tissues and cells using TRIzol reagent (Invitrogen) and 1 μg RNA was reverse transcribed to complementary DNA (cDNA) using ReverTra Ace® qPCR RT Master Mix with gDNA Remover (TOYOBO, Osaka, Japan) according to the manufacturer's instructions. Quantitative real-time PCR analyses were performed using 2× SYBR Green qPCR Master Mix (Bimake, Shanghai, CN) and specific primers on a LightCycler 480 (Roche, Basel, Switzerland). All sequences for qRT-PCR used in this study are listed in Supplementary table 5.

### Cell proliferation and colony formation assays
For cell proliferation assays, a total of 15,000 cells were seeded into 12-well plates. Cells were imaged every 2 h using the Incucyte S3 live-cell analysis system (Sartorius, Goettingen, Germany) for about 4 days. Images were analyzed and data were generated by Incucyte 2019B Rev2 software. For colony formation assays, 500 cells were seeded in the 6-well plates and incubated with normal medium for 7–10 days. Clones were fixed in 4% paraformaldehyde for 10 min followed by

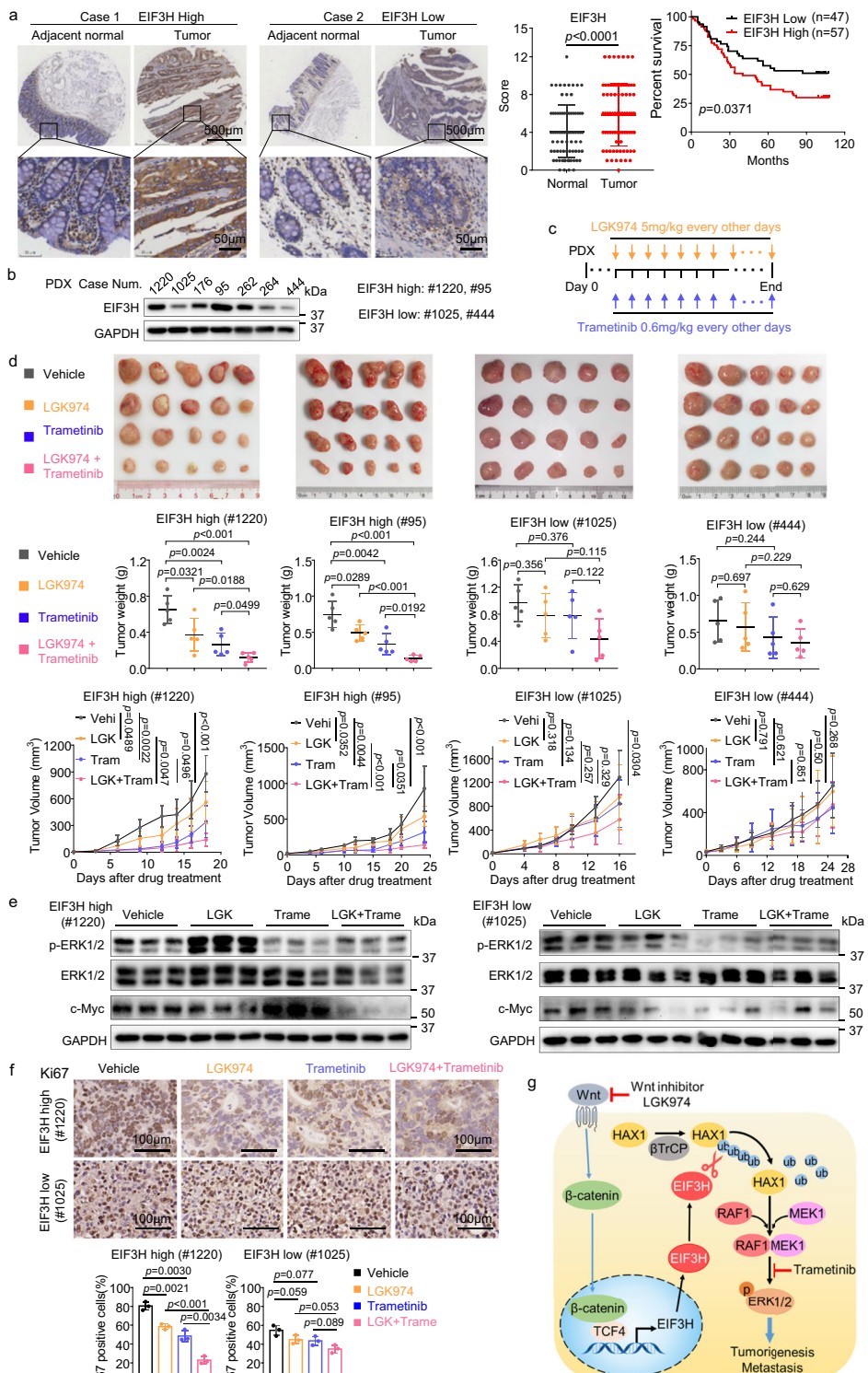

**Fig. 7 | Combined Wnt and RAF1-ERK1/2 signaling inhibitors treatment suppresses tumor growth in EIF3H-high PDX with better efficacy. a** Representative IHC staining images showing high and low expression of EIF3H in human CRC and adjacent non-tumor tissue from tissue microarray (TMA) (left). EIF3H protein levels in 76 paired human CRC and the non-tumor tissues. Data were presented as means ± SD. The *p* value was based on the two-sided Wilcoxon test (middle). The correlation between EIF3H protein level and the overall survival of CRC patients (n = 104) was tested by Kaplan-Meier analysis (right). The *p* was based on the log-rank test. **b** Immunoblot images showing EIF3H expression in seven cases of PDX tumors. **c** Schematic diagram of the LGK974 and Trametinib combination treatment after the establishment of PDX tumors in NCG mice. **d** Representative images (top) and weight (middle) of the PDX tumors that were harvested at the end of the

experiment. Growth curves showing the proliferation of PDX tumors in each indicated treatment group (bottom). n = 5 biological replicates. Data were presented as means ± SD. The *p* values were calculated by two-way ANOVA. **e** Immunoblot images showing the indicated protein levels analyzed from the PDX tumors that were treated with LGK974 and Trametinib. **f** Representative images (top) and quantifications (bottom) of IHC staining for Ki67 in PDX tumors from the indicated treatment groups. Data were presented as means ± SD, n = 3 biologically independent experiments. The *p* values were calculated by one-way ANOVA. **g** Graphical summary of the key findings of the study. Representative immunoblots shown in figures were repeated three times independently with similar results. Source data are provided as a Source Data file. PDX patient-derived xenografts.

staining with 0.5% crystal violet at room temperature for 10 min, and colony ( > 50 cells) numbers were counted.

## Transwell assays
Transwell migration and invasion assays were performed using Transwell chambers with filter membranes of 8-μm pore size (Corning Costar, New York, USA). The Transwell chamber used for invasion tests was pre-coated with 100 μL of BD Matrigel™ matrix (1:6 dilution) for 30 min at 37 °C. A total of $2 \times 10^5$ cells suspended in serum-free medium were added to the upper chamber, while 600 μl of medium containing 10% FBS was added to the lower chamber. Later 24–30 h, the cells were fixed with 4% paraformaldehyde for 30 min and stained with 0.5% crystal violet solution for 15 min. Cells were counted under a light microscope. All experiments were conducted three times in triplicate.

## Chromatin immunoprecipitation (ChIP) assays
ChIP assays were performed as described (Millipore, Catalogue # 17–10085). Briefly, intracellular protein-DNA complexes from HCT116 cells were cross-linked with 1% formaldehyde, sonicated, and subjected to chromatin-conjugated IP using specific antibodies. β-catenin (5 μg per reaction, BD, 610153), TCF4 (5 μg per reaction, Santa Cruz, sc-166699) and IgG (5 μg per reaction; Millipore, #12–370) were used in this assay. Then the precipitated DNA were purified and analyzed by qPCR with the primers specific for EIF3H promoter shown in Supplementary Table 6.

## Colorectal cancer patient-derived xenograft and orthotopic mouse models
For the orthotopic mouse model, 5–6 weeks female NCG mice (Strain NO.T001475, GemPharmatech Co., Ltd, China) were anesthetized with 2% isoflurane, and the cecum was exposed through a lower-abdomen incision. $1 \times 10^6$ luciferase-expressing HCT116 cells, containing stably tetracycline-inducible shEIF3H with or without HAX1 overexpression, were injected subserosally at the cecum wall. The mice in EIF3H KD group were intraperitoneally injected with 50 mg/kg doxycycline every two days. At the end of the study (Day 23), the tumor burden was monitored IVIS Spectrum In Vivo Imaging System upon intraperitoneally injection of D-luciferin (150 mg/kg per mouse). The mice were sacrificed and the intestines, livers and lungs were isolated and quantified with an IVIS Spectrum. Quantifications were made with Living Image v.4.5.2. The animal procedures were approved by the Animal Ethical and Welfare Committee of The Sixth Affiliated Hospital of Sun Yat-sen University (Ethical code: IACUC-2020122903, IACUC-2022062201).

To generate CRC PDX models, fresh tumor samples from CRC patients were subcutaneously implanted into 5–6 weeks old female NCG mice (Strain NO.T001475, GemPharmatech Co., Ltd, China). When the established PDXs size reached 1500 mm³, the tumors were divided into equal volume ~2 mm³ and were subcutaneously transplanted into the flanks of 5–6 weeks old female NCG mice. When tumor volume reached ~50 mm³, all mice were randomized into the following treatment groups: (1) vehicle control; (2) LGK974 (5 mg/kg, Apexbio, B2307), (3) Trametinib (0.5 mg/kg, MedChemExpress, HY-10999); (4) LGK974 and Trametinib. LGK297 and Trametinib were administered to mice via intraperitoneal injection every other day. Mice tumor volumes and weights were measured twice per week during the experiments. The maximal tumor burden permitted by ethics committee is no more than 1500 mm³. When tumor burden reached 1500 mm³, mice were euthanized and the tumors were dissected for further analysis.

## Bioinformatics analysis
GEO databases (GSE5204, GSE41258, GSE8671 and GSE77434) were accessed and analyzed by online tool GEO2R. EIF3H mRNA levels in the colon adenocarcinoma (COAD), rectal adenocarcinoma (READ) and normal tissues from the TCGA database, and EIF3H protein level in normal and colon cancer tissues from the Clinical Proteomic Tumor Analysis Consortium (CPTAC) database, was analyzed through an online tool (http://ualcan.path.uab.edu/index.html). TCGA PanCancer Proteomic data, including breast cancer ($n=105$), colorectal cancer ($n=90$), and ovarian cancer ($n=121$), were downloaded from the cBioportal dataset (https://www.cbioportal.org/datasets). Analyzed the proteomics data and the protein correlation with Z-score using Pearson's correlation (Graphpad Prism 7). Proteomics data of CPTAC colon cancer cohort were analyzed by online tool (https://cprosite.ccr.cancer.gov/). The mNRA Correlation data were also derived from online tool (https://cprosite.ccr.cancer.gov/).

## Statistics and reproducibility
All data shown (including immunoblotting, immunofluorescent staining) were obtained from at least three biological independent experiments with similar results. Data were analyzed by Student's $t$ test, one-way or two-way Analysis of Variance (ANOVA) using GraphPad Prism 7. To evaluate EIF3H mRNA expression in CRC patients' tumor tissues, the sample size is determined to have 90% power to identify differences of 1.4 SDs between the two groups at a significance level of $\alpha = 0.05$. Cumulative survival was evaluated using the Kaplan-Meier method (log-rank test). Cox proportional hazards regression for univariate and multivariate analyses were employed to evaluate which clinicopathologic factors had prognostic values. For animal studies, no statistical method was used to predetermine sample size.

## Reporting summary
Further information on research design is available in the Nature Portfolio Reporting Summary linked to this article.

## Data availability
The data that support the findings of this study are available within the article and its Supplementary Information files. The mass spectrometry proteomics data are deposited in ProteomeXchange with identifier PXD048151 and PXD048485(https://proteomecentral.proteomexchange.org/) via iProX partner repository (IPX0007834000, IPX0007842000, https://www.iprox.cn/page/home.html). Source data are provided in this paper.

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

## Acknowledgements

This work was supported by the National Key R&D Program of China (2020YFA0803300 to M.L.), National Key Clinical Discipline, the National Natural Science Foundation of China (82273133 to X.M. and 81630072 to M.L.), Guangdong Basic and Applied Basic Research Foundation (2023A1515030261 and 2021A1515012081 to X.M.), the Open Fund of Guangdong Provincial Key Laboratory of Colorectal and Pelvic Floor Diseases (2020B1212060023 to R.L.), the Guangzhou Science and Technology Program Project (202206010167 to M.L.).

## Author contributions

M.L. and X.M. conceived the project. H.J. designed and participated in most of the experiments with the help of X.C., B.Z., N.M. and Y.W. X.H. performed the in vivo animal work, IHC staining, and IF experiments. Q.P. performed the bioinformatics analysis and ChIP experiments. F.Y. and W.W. established the PDX model. X.X. and N.M. cloned plasmids. P.Z., J.W., J.L., and R.L. made critical advices, comments, and assis-tance in the study. H.J. performed data analysis and drafted the manuscript. M.L. and X.M. supervised the project and revised the manuscript. All authors reviewed and approved the manuscript for publication.

## Competing interests

The authors declare no competing interests.
