## [Peer Review File · Nature Communications]

REVIEWER COMMENTS

Reviewer #1 (Remarks to the Author):

In this manuscript Jin and colleagues propose that an EIF3H-HAX1 axis promotes RAF-MEK-ERK signaling and colorectal cancer growth/metastasis. The authors show that EIF3H is overexpressed in human colorectal cancer and that EIF3H deletion in mice reduces colitis-induced colorectal tumorigenesis. They also show that EIF3H induces human CRC cell growth, migration and invasion. They also demonstrate that HAX1 is targeted for degradation by the SCFbetaTrCP ubiquitin ligase complex and that HAX1 expression results in increased interaction between RAF1, MEK1 and ERK1 thus leading to ERK1/2 phosphorylation and activation. Finally, it is shown that the EIF3H/HAX1 axis promotes CRC tumorigenesis and metastasis in a mouse orthotopic cancer model and that targeting both Wnt and RAF1-ERK1/2 signaling pathways synergistically decreases tumor growth in EIF3H-high patient-derived xenografts.

In my opinion the molecular mechanisms proposed in this study are not supported by the data presented.

The data demonstrating that HAX1 is a substrate of the SCFbetaTrCP ubiquitin ligase complex are preliminary.

1. Several previous studies have demonstrated that HAX1 interacts with the F-box protein FBXO25 (doi: 10.1002/jcp.30044; doi: 10.1038/nm.3740; doi: 10.1074/jbc.RA120.014616). The authors should test the specificity of the HAX1-betaTrCP binding by immunoprecipitating a panel of at least 5-6 F-box proteins including FBXO25, betaTrCP1 and its paralog betaTrCP2 and test their interaction with endogenous HAX1.

2. The putative betaTrCP-binding domain (phosphodegron) in HAX1 is not canonical – it does not contain the glycine as in established betaTrCP substrates. More importantly, the sequence proposed as phosphodegron is not conserved in other mammals (mouse and rat) as the threonine residue is missing.

3. Have the authors tested whether silencing of both betaTrCP1 and 2 results in HAX1 stabilization by CHX chase?

4. Does overexpression of betaTrCP-deltaF-box induces HAX1 accumulation?

5. Figure 3E: the effect of betaTrCP overexpression on HAX1 stability is not clear.

Overall, the data supporting the model that HAX1 works as a scaffold protein enhancing the interaction between RAF1, MEK1 and ERK1, and potentiating ERK1/2 phosphorylation and activation are not conclusive. For instance, in Figure 4B there is no correlation between the efficiency of HAX1 silencing and the effect on ERK1/2 phosphorylation. Although shHAX1 #1 is more efficient than shHAX1 #2 in knocking down HAX1 in DLD1 and HCT116 cells, it has less effect on the phosphorylation of ERK1/2.

Reviewer #2 (Remarks to the Author):

In the study by Jin et al., the authors show that EIF3H expression is increased in colorectal cancer. Using colitis-induced CRC mouse models, heterozygous deletion of Eif3h resulted in a reduced tumor load and size. Furthermore, KD of EIF3H could reduce tumor proliferation and invasion/migration in CRC cell lines. On a mechanistic levels, the authors propose that EIF3H directly interacts with HAX1, affecting its protein stability via deubiquitination and counteracting its ubiquitination by β -TRCP. They found that HAX1 enhances the interaction of RAF/MEK/ERK, resulting in increased phospho-ERK levels, and thereby mediates the effect of EIF3H. Finally, the authors show that combined Wnt/MEK inhibition effectively reduces PDX growth in EIF3H high tumors.

The results are extensive and interesting, and the experiments are well designed. There are a few open questions that should be addressed in a revised version.

Major:

1. The authors describe that Eif3h flox/flox X Villin-CreERT are embryonic lethal and that this is the result of complete loss-of-function. However, no tamoxifen was added, so in principle, no recombination should have happened. To assess the leakiness of the system, I would recommend to compare Eif3h fl/wt X Villin-CreERT +/- TAM. The knockout itself seems to be mild as shown in Figure 1D, and I would suggest to use a second method (immunoblot) to quantify protein levels.
2. If EIF3H directly regulates HAX1 levels, there should be a difference in HAX1 protein/transcript levels in EIF3H high/low tumors. The authors could re-analyse publicly available datasets to validate this correlation. In this regard, it is also interesting if the genetic status of the tumors affect EIF3H levels, as the authors suggest that high Wnt activity (for instance mediated by APC mutations) could result in higher EIF3H levels.
3. Since EIF3H is part of the eIF3 complex, any interference/depletion would result in a global change of translation, thereby affecting many cellular processes. How these global effects compare to the specific effect on HAX1 is a key question. The authors could assess global protein abundances via MS after EIF3H KD to provide an insight into this question.
4. The authors used CMYC as a readout for the Wnt pathway, I would recommend to use AXIN2, as CMYC can be influenced by activity of the RAS-MAPK pathway as well. Also NCB0846 is not a well recognized Wnt inhibitor. Using ICG-001 or tankyrase inhibitors would be better. Maybe the authors can add one set of experiments with one of the two inhibitors.

Minor:

1. In Figure 6J, the concentrations used for Trametinib seem to be extremely high for such a potent inhibitor. Did the authors confuse μM and nM ?
2. In some sections, the language could be improved and spelling mistakes should be avoided.

Reviewer #3 (Remarks to the Author):

The amplification of eukaryotic initiation translation factor 3 subunit h (EIF3H) has been found in various cancers. However, whether and how EIF3H contributes to tumorigenesis remains ambiguous. In this study, Jin et al. employed AOM/DSS induced colorectal cancer model to determine the role of EIF3H in cancer development since it was remarkably up-regulated in human CRC samples. They found that depleting EIF3H in intestinal epithelial cells significantly impaired colitis-induced colorectal tumorigenesis. Mechanistically, they further demonstrated that EIF3H functioned as a deubiquitinase to antagonize the ubiquitination of HAX1 by βTrCP , and hence impaired the degradation of HAX1. Next, they found that HAX1 interacted with RAF1 and activated RAF/MEK/ERK signaling while active Wnt/ β -catenin signaling promotes the expression of EIF3H in colorectal cancers, which constructs a signaling axis of Wnt/ β -catenin-EIF3H-HAX1-RAF/MEK/ERK that is responsible for colorectal cancer development. To validate this finding and translate it to clinic cancer treatment, authors determined whether blocking both Wnt/ β -catenin signaling and ERK signaling could effectively inhibit the growth of colorectal cancer with high EIF3H expression by using PDX model, and indeed they found this combination exhibited an excellent efficacy. Overall, this is an excellent study with compelling experimental data, which would have important implications in colorectal cancer treatment. Authors could address my following comments to improve their manuscript.

Major points:

Although overall authors provided pretty strong data to support their conclusion, how HAX1 binds to and activates RAF1 is ambiguous. Does HAX1 association promote RAF1 homodimerization or heterodimerization with BRAF since RAF dimerization is critical for its activation as well as substrate phosphorylation? Does HAX1 also interact with the other two RAF isoforms, ARAF and BRAF since they have quite similar structures, particularly the N-lobe of kinase domain that mediates HAX1-RAF1

interaction? In addition, the NTA motif phosphorylation of RAF1 is a marker for RAF1 activation, does co-expression of HAX1 with RAF1 trigger this event? If authors can address these questions, this manuscript will be greatly improved.

Minor points:

1. There are pretty much English grammar errors in the whole manuscript. Please check the manuscript carefully and correct spelling/grammar errors. For example, line 104, Eif3hfl/oxwt should be Eif3hflox/wt.
2. Images in Figure 1D are not clear. Authors should provide high-resolution images.
3. Figure 3B, the anti- β TrCP immunoblot for the last co-immunoprecipitation assay, it's better for authors to do this immunoblot with the same membrane for anti-HAX1 immunoblot.

Response to Reviewers

Reviewer #1 (Remarks to the Author):

In this manuscript Jin and colleagues propose that an EIF3H-HAX1 axis promotes RAF-MEK-ERK signaling and colorectal cancer growth/metastasis. The authors show that EIF3H is overexpressed in human colorectal cancer and that EIF3H deletion in mice reduces colitis-induced colorectal tumorigenesis. They also show that EIF3H induces human CRC cell growth, migration and invasion. They also demonstrate that HAX1 is targeted for degradation by the SCFbetaTrCP ubiquitin ligase complex and that HAX1 expression results in increased interaction between RAF1, MEK1 and ERK1 thus leading to ERK1/2 phosphorylation and activation. Finally, it is shown that the EIF3H/HAX1 axis promotes CRC tumorigenesis and metastasis in a mouse orthotopic cancer model and that targeting both Wnt and RAF1-ERK1/2 signaling pathways synergistically decreases tumor growth in EIF3H-high patient-derived xenografts.

In my opinion the molecular mechanisms proposed in this study are not supported by the data presented.

The data demonstrating that HAX1 is a substrate of the SCFbetaTrCP ubiquitin ligase complex are preliminary.

1. Several previous studies have demonstrated that HAX1 interacts with the F-box protein FBXO25 (doi: 10.1002/jcp.30044; doi: 10.1038/nm.3740; doi: 10.1074/jbc.RA120.014616). The authors should test the specificity of the HAX1-betaTrCP binding by immunoprecipitating a panel of at least 5-6 F-box proteins including FBXO25, betaTrCP1 and its paralog betaTrCP2 and test their interaction with endogenous HAX1.

Response: We thank the reviewer for the questions and suggestions. We have constructed 5 F-box proteins' plasmids with Flag-tag or HA-tag including β TrCP1, β TrCP2, FBXO25, FBXO1 and SKP2. Then we transfected those plasmids into HEK293T cells and performed exogenous co-IP assay (Response Figure 1 A-E). Our results showed that FBXO25, β TrCP1 and its paralog β TrCP2 could interact with endogenous HAX1, while the other two F-box proteins, SKP2 and FBXO1, could not. This was consistent with our mass spectrometry results, as no unique peptide corresponding to SKP2 or FBXO1 was identified. Co-IP experiments also showed that HAX1 still associated with F-box deleted mutant β TrCP1 (a ubiquitination deficient mutant in which the substrate binding moiety is intact) (Response Figure 1F).

Response Figure 1. The interaction between exogenous F-box proteins and endogenous HAX1 was determined by co-IP assay. HEK293T cells were transfected with Flag-F-box protein or HA-FBXO25 plasmid. The cell lysates from HEK293T transfected with Flag-F-box protein plasmid were pulled down with anti-Flag M2 beads and immunoblotted with the indicated antibodies. Cell lysates from HEK293T transfected with HA-FBXO25 plasmid were immunoprecipitated with either control rabbit IgG or HA antibodies followed by immunoblotting with indicated antibodies.

2. The putative betaTrCP-binding domain (phosphodegrom) in HAX1 is not canonical – it does not contain the glycine as in established betaTrCP substrates. More importantly, the sequence proposed as phosphodegrom is not conserved in other mammals (mouse and rat) as the threonine residue is missing.

Response: We thank the reviewer for the questions. βTrCP recognizes a DSGXXS motif or its variants (for example, DSG/DDG/EEG/SSGXXS/E/D motifs) in which the serine residues are phosphorylated by specific kinases to allow binding to βTrCP (PMID: 18500245). Here we identified one noncanonical βTrCP binding degrom motif (231 DSEGRT 236) in HAX1, which meets the requirement of two positively charged/phosphorylated residues for binding βTrCP. It has been reported that some proteins have no-glycine degrom motif that interacts with βTrCP for subsequent ubiquitination and degradation, such as WEE1, p100, CDC25A, MCL1 and TIPE2 (PMID: 15070733, 11994270, 14603323, 17387146, 32188758, 19797085). We think our HAX1 belongs to this category.

Although the sequence proposed as phosphodegrom is not conserved in *Mus musculus* and *Rattus rattus* (rodents), it is indeed conserved in most other mammals, including *Pan troglodytes*, *Callithrix jacchus*, *Macaca mulatta* (primates); *pteropus alecto* (chiroptera); *Canis lupus familiaris*, *Ursus maritimus*, *Eumetopias jubatus* (carnivora); *Erinaceus europaeus* (insectivores); *Equus caballus* (odd-toed ungulates); *Manis pentadactyla* (pangolins)

(<https://www.ncbi.nlm.nih.gov/tools/cobalt/cobalt.cgi>).

Some reported substrates of β -TrCP		HAX1 phosphodegron in some Mammals/Eutheria				
β -TrCP substrates	Degron	Mammals/Eutheria	Species	Start	Alignment	end
I κ B α	DSGLDS	primates	Homo sapiens	228	TVVDSEGR I ETTV	240
Snail	DSGKGS	primates	Pan troglodytes	228	TVVDSEGR I ETTV	240
ATF4	DSGICMS	primates	Macaca mulatta	228	TVVDSEGR I ETTV	240
WEE1	DSAFQE	primates	Callithrix jacchus	232	TVVDSEGR I ETTV	244
p100	DSAYGS	chiroptera	Pteropus alecto	227	TVVDSEGR I ETTV	239
CDC25A	STDSG	carnivores	Canis lupus familiaris	242	TVVDSEGR I ETTV	254
MCL1	DGSLPS	carnivores	Ursus maritimus	229	TVVDSEGR I ETTV	241
PHLPP1	QSVLLT/LSVEE	carnivores	Eumetopias jubatus	235	TVVDSEGR I ETTV	247
TIPE2	MESFSSKS	insectivores	Erinaceus europaeus	227	TVVDSAGR I ETTV	239
		odd-toed ungulates	Equus caballus	226	TVVDSEGR I ETTV	238
		pangolins	Manis pentadactyla	228	TVVDSEGR I ETTV	240
		rodents	Mus musculus	229	TVVDSEGR R ETTV	241
		rodents	Rattus rattus	227	TVVDSEGR R ETTV	239

3. Have the authors tested whether silencing of both betaTrCP1 and 2 results in HAX1 stabilization by CHX chase?

Response: We thank the reviewer for the suggestion. We used siRNA to silence β TrCP1 and β TrCP2. The silencing efficiency was examined by western blot (β TrCP1) or qRT-PCR (β TrCP2). We observed that silencing of both β TrCP and β TrCP2 resulted in HAX1 stabilization in CRC cells by CHX chase assay (Response Figure 3, also added to Revised Supplementary Fig. 3d).

Response Figure 3. HAX1 turnover rate was analyzed by CHX assay in HCT116 cells transfected with siRNA targeting β TrCP or β TrCP2.

4. Does overexpression of betaTrCP-deltaF-box induces HAX1 accumulation?

Response: We thank the reviewer for the question. Yes, the expression of HAX1 were not affected by overexpression of delta F-box β TrCP in the HCT116 cells (Response Figure 4, also added to Revised Fig. 3f), as delta F-box β TrCP was a ubiquitination deficient mutant.

Response Figure 4. Representative immunoblots showing HAX1 steady-state expression in HCT116 cells upon delta F-box β TrCP overexpression.

5. Figure 3E: the effect of betaTrCP overexpression on HAX1 stability is not clear.

Response: Thanks for the concern. We re-performed this CHX chase assay and replaced the less clear image for revised version (Response Figure 5, also added to Revised Fig. 3e). As showed in this figure, β TrCP overexpression led to accelerated turnover rate of HAX1, while the F-box mutant β TrCP had no effect.

Response Figure 5. HCT116 cells transfected with either WT Flag- β TrCP or F-box mutant β TrCP were treated with CHX for the indicated times. The cell lysates were immunoblotted with indicated antibodies (left). The turnover rate of HAX1 was shown (right).

Overall, the data supporting the model that HAX1 works as a scaffold protein enhancing the interaction between RAF1, MEK1 and ERK1, and potentiating ERK1/2 phosphorylation and

activation are not conclusive. For instance, in Figure 4B there is no correlation between the efficiency of HAX1 silencing and the effect on ERK1/2 phosphorylation. Although shHAX1 #1 is more efficient than shHAX1 #2 in knocking down HAX1 in DLD1 and HCT116 cells, it has less effect on the phosphorylation of ERK1/2.

Response: Thanks a lot for reviewer insightful questions and constructive suggestions. To add more evidence, we used one more HAX1 silencing sequence (totally 3 shRNA sequence) to re-performed western blot analysis in two CRC cell lines. Based on the new results, we think there is a correlation between ERK1/2 phosphorylation and HAX1 silencing efficiency (Response Figure 6A, also added to Revised Fig. 4b). More importantly, we also performed co-IP experiments while increasing HAX1 expression. The results showed that increasing HAX1 expression enhanced the RAF1 homodimerization and heterodimerization with BRAF (Response Figure 6B-C, also added to Revised Fig. 4g-h). These results suggest that HAX1 could promote ERK1/2 phosphorylation level by enhancing the RAF1 homodimerization and heterodimerization with BRAF.

Response Figure 6. (A) Effect of HAX1 knockdown on the phosphorylation of ERK1/2 in DLD1 and HCT116 cells was detected by western blotting. (B-C) The HEK293T cells were transfected with Myc-RAF1, Flag-BRAF (B) or Flag-RAF1 (C) with increasing HA-HAX1 plasmids. Cell lysates were immunoprecipitated with anti-Myc beads and immunoblotted with indicated antibodies. WCL, whole cell lysis.

Reviewer #2 (Remarks to the Author):

In the study by Jin et al., the authors show that EIF3H expression is increased in colorectal cancer. Using colitis-induced CRC mouse models, heterozygous deletion of *Eif3h* resulted in a reduced tumor load and size. Furthermore, KD of EIF3H could reduce tumor proliferation and invasion/migration in CRC cell lines. On a mechanistic levels, the authors propose that EIF3H directly interacts with HAX1, affecting its protein stability via deubiquitination and counteracting its ubiquitination by β -TRCP. They found that HAX1 enhances the interaction of RAF/MEK/ERK, resulting in increased phospho-ERK levels, and thereby mediates the effect of EIF3H. Finally, the authors show that combined Wnt/MEK inhibition effectively reduces PDX growth in EIF3H high tumors.

The results are extensive and interesting, and the experiments are well designed. There are a few open questions that should be addressed in a revised version.

Major:

1. The authors describe that *Eif3h* flox/flox X Villin-CreERT are embryonic lethal and that this is the result of complete loss-of-function. However, no tamoxifen was added, so in principle, no recombination should have happened. To assess the leakiness of the system, I would recommend to compare *Eif3h* fl/wt X Villin-CreERT +/- TAM. The knockout itself seems to be mild as shown in Figure 1D, and I would suggest to use a second method (immunoblot) to quantify protein levels.

Response: Thanks a lot for the reviewer's constructive comments. The purpose of using this mouse model is to study if knockout or knockdown of EIF3H has any effect on colon tumorigenesis. We believe there is a partial leakiness of *Villin*^{CreERT}, since the embryonic lethal happened even without tamoxifen injection. That's why we choose the partial knock out model. Two partial knock out strategy (①: *Eif3h*^{flox/wt} VS *Eif3h*^{flox/wt} *Villin*^{CreERT}, **ALL inject TAM**; ②: *Eif3h*^{flox/wt} *Villin*^{CreERT} **-TAM** VS *Eif3h*^{flox/wt} *Villin*^{CreERT} **+TAM**) both make sense. We agree that strategy ② really can assess the leakiness of the system. It is warranted to be investigated in the future. The reasons why we choose "①" are that: (1) To achieve our purpose, strategy "①"

will maximize *Eif3h* expression difference between wt and partial KO group since the leakiness happen (The partial knockout efficiency was confirmed in the following Response Figure 7A-C by IF, WB and IHC). (2) To avoid TAM effect and reduce breeding times. The disadvantage of “①” strategy is that partial leakiness of *Villin*^{CreERT} in other organ may have influence on mice development to further affect our conclusions. In fact, we checked *Eif3h* heterozygous knockout mice (*Eif3h*^{fl/wt} *Villin*^{CreERT} VS *Eif3h*^{fl/wt}) during the development. Our data showed the partial knockout mice have normal development with no overt phenotype in the gut, normal mating and breeding, compared to their *Eif3h*^{fl/wt} littermates (Revised Supplementary Fig. 2b and c).

We apologize that we did not present the knockout efficiency clearly. We relabeled the immunofluorescence (IF) data and showed that representative images of IF of colon tissues obtained from the indicated tamoxifen-induced mice (Response Figure 7A, Revised Fig. 1d).

. Red arrows indicate the cells expressing CRE-ERT have less expression of *Eif3h* (purple) in *Eif3h*^{-/wt} mouse (*Eif3h*^{fl/wt} *Villin*^{CreERT} treated with TAM) when compared with *Eif3h*^{fl/wt} mice. As suggested by the reviewer, we also performed immunoblot and immunohistochemistry staining assays to show that the protein level of *Eif3h* was indeed decreased in *Eif3h* heterozygous knockout mice (Response Figure 7B and C, also added to Revised Supplementary Fig. 2d and 2e).

Response Figure 7. (A) Representative images of IF staining of colon tissues obtained from the indicated tamoxifen-induced mice. Red arrows indicated the cells (expressing CRE-ERT and having recombination happen) that had less expression of *Eif3h* (purple) in *Eif3h*^{-/wt} mouse when compared with *Eif3h*^{fl/wt} mice. Nuclei stained with DAPI (blue). Scale bar = 20µm. (B)

Protein levels of EIF3H in indicated mouse colon tissues. (C) Representative images of Immunohistochemistry staining of colon tissues obtained from the indicated tamoxifen-induced mice. Scale bar = 25 μ m.

2. If EIF3H directly regulates HAX1 levels, there should be a difference in HAX1 protein/transcript levels in EIF3H high/low tumors. The authors could re-analyse publicly available datasets to validate this correlation. In this regard, it is also interesting if the genetic status of the tumors affect EIF3H levels, as the authors suggest that high Wnt activity (for instance mediated by APC mutations) could result in higher EIF3H levels.

Response: Thanks a lot for the reviewer's comments. We analyzed CPTAC colon cancer cohort and TCGA-CRC cohort, and found that the HAX1 positive correlated with EIF3H expression in CPTAC colon cancer cohort (<https://cprosite.ccr.cancer.gov/>) (Response Figure 8A-B, also added to Revised Supplementary Figure 4a). We also analyzed changes in mRNA levels of *Eif3h* from datasets of colon tumors isolated from *Apc*^{Min/+} and AOM-treated mice (GSE5204), and found that *Eif3h* mRNA levels were elevated in tumors of *Apc*^{Min/+} and AOM-induced mice compared with normal mice colon tissues (Response Figure 8C, Revised Supplementary Fig. 8a). In TCGA database, EIF3H expression was positively correlated with the mRNA levels of AXIN2 and MYC, which are downstream genes of Wnt pathway (Response Figure 8D, also added to Revised Supplementary Fig. 8b).

Response Figure 8. (A-B) Spearman rank correlation analysis of EIF3H and HAX1 protein expression in CRC tumors from TCGA-CRC cohort and CPTAC colon cancer cohort. (C) Using GEO2R of PubMed (<http://www.ncbi.nlm.nih.gov/geo/geo2r/>), we assessed changes in *Eif3h* mRNA levels in Gene Expression Omnibus datasets of colon tumors isolated from *Apc*^{Min/+} and AOM-treated mice (GSE5204). (D) Spearman rank correlation analysis of EIF3H and AXIN2, MYC mRNA level in CRC tumors from TCGA-CRC cohort.

3. Since EIF3H is part of the eIF3 complex, any interference/depletion would result in a global change of translation, thereby affecting many cellular processes. How these global effects compare to the specific effect on HAX1 is a key question. The authors could assess global protein abundances via MS after EIF3H KD to provide an insight into this question.

Response: Thanks for the reviewer's question and suggestion. First, we measured the nascent protein synthesis in HCT116 with/without induction of EIF3H KD using the Click-iT® Plus OPP Alexa Fluor® 488 Protein Synthesis Assay Kit (PMID: 26319900). EIF3H KD resulted in partial inhibition of nascent protein synthesis compared to the control group (Response Figure.9A). As a positive control, the nascent protein synthesis is significantly decreased in Cycloheximide (CHX) treated group. Previous works have reported that overexpression of EIF3H in NIH-3T3 cells results in significant activation of protein synthesis, especially some oncogenic mRNAs involved in growth control (PMID: 17170115, 18544531, 23716667). We also measured the specific effect of EIF3H knockdown on the translational efficiencies of HAX1, CCND1 and ACTB mRNAs by polysome profiling assay. Although a shift of ribosomes toward the lightest polysomes, relative to the control cell, was observed in the EIF3H knockdown HCT116 cell (Response Figure 9C, also added to Revised Supplementary Fig. 4f), EIF3H knockdown could not reduce the polysome size of HAX1 mRNAs, as evidence by no shift of the polysome peak to smaller polysomes (Response Figure 9D, also added to Revised Supplementary Fig. 4g-h). This observation of HAX1 was similar to GAPDH mRNA reported by Zhang et.al (PMID: 17170115, PMID: 18544531). ACTB and CCND1 act as a negative and positive control (PMID: 18544531). Taken together, these results suggested that EIF3H knockdown has no effect on HAX1 protein synthesis, although EIF3H affects global nascent protein synthesis. We have added the results to Revised Supplementary Figure 4f-h, and the method to supplemental materials and methods.

Response Figure 9. (A-B) Evaluation of EIF3H KD on nascent protein synthesis. Two days after doxycycline induction, O-propargyl-puromycin (OPP) was added to the culture medium to label nascent peptides which were visualized by Immunofluorescence microscope after fixation with fluorescent Click iT chemistry. The fluorescence intensity of all the cells corresponding to protein synthesis. Representative images of protein synthesis were shown (A). Data represented means \pm SD from three independent experiments (B). (C) Polysome profiles of EIF3H KD and control cells. HCT116 Cells were knocked down of EIF3H, further lysed and subjected to sucrose gradient centrifugation. The sucrose gradient profiles were obtained by continuous scanning at A254. The positions in the gradients of 40S subunits, 80S ribosomes, and polysomes were labeled. The translational status of indicated mRNA in EIF3H KD and control cells were examined by qPCR (D). EIF3H knockdown had no effect on HAX1 protein synthesis.

4. The authors used CMYC as a readout for the Wnt pathway, I would recommend to use AXIN2, as CMYC can be influenced by activity of the RAS-MAPK pathway as well. Also NCB0846 is not a well recognized Wnt inhibitor. Using ICG-001 or tankyrase inhibitors would be better. Maybe the authors can add one set of experiments with one of the two inhibitors.

Response: Thanks a lot for constructive suggestions. As suggested by the reviewer, we used ICG-001 in different concentration to treat CRC cells and checked the change of EIF3H mRNA/protein level. Treatment with ICG-001 reduced the expression of EIF3H at both mRNA

and protein levels in a dose-dependent manner (Response Figure 10A-B). We also used AXIN2 as a readout for the Wnt pathway to re-analyze the mRNA samples from DLD1, HCT116 and RKO cells treated with Wnt pathway inhibitor NCB0846. The results showed that both EIF3H and AXIN2 transcription could be inhibited by NCB0846 (Response Figure 11A-B). All those results were added to Revised Fig. 6b-c, and Revised Supplementary Fig. 8c.

Response Figure 10. qRT-PCR (A) and western-blot (B) analysis of EIF3H and AXIN2 level in DLD1 and HCT116 cells treated with different concentrations of Wnt pathway inhibitor ICG-001.

Response Figure 11. qRT-PCR (A) and western-blot (B) analysis of EIF3H and AXIN2 levels in DLD1, HCT116 and RKO cells treated with different concentrations of Wnt pathway inhibitor NCB0846. Both EIF3H and AXIN2 transcription could be inhibited by NCB0846.

Minor:

1. In Figure 6J, the concentrations used for Trametinib seem to be extremely high for such a

potent inhibitor. Did the authors confuse μM and nM ?

Response: We thank the reviewer to point out this mistake. In Figure 6J, the concentrations used for Trametinib was nM . We have corrected this mistake.

2. In some sections, the language could be improved and spelling mistakes should be avoided.

Response: Thank a lot for reviewer's kind reminder. We apologize for spelling and grammatical mistakes and have corrected them throughout the text accordingly.

Reviewer #3 (Remarks to the Author):

The amplification of eukaryotic initiation translation factor 3 subunit h (EIF3H) has been found in various cancers. However, whether and how EIF3H contributes to tumorigenesis remains ambiguous. In this study, Jin et al. employed AOM/DSS induced colorectal cancer model to determine the role of EIF3H in cancer development since it was remarkably up-regulated in human CRC samples. They found that depleting EIF3H in intestinal epithelial cells significantly impaired colitis-induced colorectal tumorigenesis. Mechanistically, they further demonstrated that EIF3H functioned as a deubiquitinase to antagonist the ubiquitination of HAX1 by βTrCP , and hence impaired the degradation of HAX1. Next, they found that HAX1 interacted with RAF1 and activated RAF/MEK/ERK signaling while active Wnt/ β -catenin signaling promotes the expression of EIF3H in colorectal cancers, which constructs a signaling axis of Wnt/ β -catenin-EIF3H-HAX1-RAF/MEK/ERK that is responsible for colorectal cancer development. To validate this finding and translate it to clinic cancer treatment, authors determined whether blocking both Wnt/ β -catenin signaling and ERK signaling could effectively inhibit the growth of colorectal cancer with high EIF3H expression by using PDX model, and indeed they found this combination exhibited an excellent efficacy. Overall, this is an excellent study with compelling experimental data, which would have important implications in colorectal cancer treatment. Authors could address my following comments to improve their manuscript.

Major points:

Although overall authors provided pretty strong data to support their conclusion, how HAX1 binds to and activates RAF1 is ambiguous. Does HAX1 association promote RAF1 homodimerization or heterodimerization with BRAF since RAF dimerization is critical for its activation as well as substrate phosphorylation? Does HAX1 also interact with the other two RAF isoforms, ARAF and BRAF since they have quite similar structures, particularly the N-

lobe of kinase domain that mediates HAX1-RAF1 interaction? In addition, the NTA motif phosphorylation of RAF1 is a marker for RAF1 activation, does co-expression of HAX1 with RAF1 trigger this event? If authors can address these questions, this manuscript will be greatly improved.

Response: Thanks a lot for the reviewer's constructive suggestions. To test whether HAX1 promotes RAF1 homodimerization or heterodimerization with BRAF, we performed co-IP (pull-down) experiments using different tagged proteins. Cells were co-transfected with Myc-RAF1/Flag-BRAF or Flag-RAF1/Myc-RAF1 in the presence of increasing HA-tagged HAX1, then co-IP experiments were performed. Increasing amounts of HA-tagged HAX1 promotes the binding of Myc-RAF1/Flag-BRAF or Flag-RAF1/myc-RAF1. Thus, our results demonstrate that increasing HAX1 enhanced the RAF1 homodimerization and RAF1/BRAF heterodimerization (Response Figure 12 A-B, also added to Revised Fig. 4g-h). Similarly, when HAX1 was knockdown, the RAF1 homodimerization and heterodimerization with BRAF was decreased, but these phenomena can be rescued by HAX1 re-expression 1 (Response Figure 12C-D, also added to Revised Fig. 4i and Supplementary Fig. 7b).

Co-IP assay confirmed the interaction between HAX1 and the other two RAF isoforms: ARAF and BRAF (Response Figure 12E, also added to Revised Fig. 4i and Supplementary Fig. 7c).

Although ERK phosphorylation has been altered by HAX1 KD or overexpression accordingly, there was no change in NTA motif phosphorylation level of RAF1 (Phospho-Ser338) (Response Figure 12F-G, Revised Fig. 4b). Further studies are warranted for this observation.

Response Figure 12. (A-B) The HEK293T cells were transfected with Myc-RAF1, Flag-BRAF (A) or Flag-RAF1 (B) with increasing HA-HAX1 plasmids. Cell lysates were immunoprecipitated with anti-Myc beads and immunoblotted with indicated antibodies. WCL, whole cell lysis. (C-D) The shCtrl and shHAX1 HCT116 cells were transfected with Myc-RAF1, Flag-BRAF(C) or Flag-RAF1 (D) with or without HA-HAX1 plasmids. Cell lysates were immunoprecipitated with anti-Myc beads and immunoblotted with indicated antibodies. (E) HEK293T cells were transfected with Flag-ARAF (top) or Flag-BRAF (bottom) plasmids. Cell lysates were immunoprecipitated with anti-Flag M2 beads and immunoblotted with

indicated antibodies. (F-G) Effect of HAX1 knockdown (F) and overexpression (G) on the NTA motif phosphorylation of RAF1 (Phospho-Ser338) in DLD1 and HCT116 cells was detected by western blotting.

Minor points:

1. There are pretty much English grammar errors in the whole manuscript. Please check the manuscript carefully and correct spelling/grammar errors. For example, line 104, *Eif3hfl/oxwt* should be *Eif3hflox/wt*.

Response: We really appreciate the reviewer's comments and apologize for those spelling and grammar mistakes. We have now revised the whole manuscript to incorporate the reviews comments and to correct other errors.

2. Images in Figure 1D are not clear. Authors should provide high-resolution images.

Response: Thanks for the reviewer's suggestions. We have modified Figure 1D in the revised manuscript (Response Figure 13A, Revised Fig. 1d). And we have also add more experiments to support our conclusion (Response Figure 13 B-C, Revised Supplementary Fig. 2d-e).

Response Figure 13. (A) Representative images of immunofluorescence staining of colon tissues obtained from the indicated tamoxifen-induced mice. Red arrows indicated the cells (expressing CRE-ERT and having recombination happen) that had less expression of *Eif3h* (purple) in *Eif3h^{-/wt}* mouse when compared to *Eif3h^{fl/wt}* mice. Nuclei stained with DAPI (blue). Scale bar = 20µm. (B) Protein levels of EIF3H in indicated mouse colon tissues. (C)

Representative images of Immunohistochemistry staining of colon tissues obtained from the indicated tamoxifen-induced mice. Scale bar = 25 μ m.

3. Figure 3B, the anti- β TrCP immunoblot for the last co-immunoprecipitation assay, it's better for authors to do this immunoblot with the same membrane for anti-HAX1 immunoblot.

Response: Thanks for the reviewer's suggestions. We performed exogenous and endogenous co-IP assays in HEK293T and HCT116 cells as show in Response Figure 14 (Also added to Revised Fig. 3b).

Response Figure 14. The interaction between exogenous and endogenous HAX1 and β TrCP was determined by co-IP assay. HEK293T cells were transfected with Flag-HAX1 or Flag- β TrCP plasmid. The cell lysates were pulled down with anti-Flag M2 beads and immunoblotted with the indicated antibodies. HCT116 cell lysates were immunoprecipitated with either control rabbit IgG, HAX1 or β TrCP antibodies followed by immunoblotting with indicated antibodies.

REVIEWERS' COMMENTS

Reviewer #1 (Remarks to the Author):

The authors have addressed most of my concerns and strengthened the manuscript.

Reviewer #2 (Remarks to the Author):

The authors have addressed my comments with additional bioinformatic analysis and novel experimental data. The results of these experiments overall support the original hypothesis.

I have only a minor comment with regard to the bioinformatic analysis: it seems strange that the TCGA-CRC cohort contains only few number of cases as indicated in the response Figure 8A. The authors should indicate which specific TCGA dataset is used as there are several. I recommend to reanalyze with the TCGA Pancreas Cancer dataset. Furthermore, the analysis should be clearly described in the methods section.

Overall, the manuscript has been significantly improved and once the minor comment has been addressed, it is suitable for publication.

Reviewer #3 (Remarks to the Author):

Authors has addressed my comments, which significantly improves this manuscript. I have no further comments and support its publishing in Nature Communication.

Response to Reviewers

Reviewer #2 (Remarks to the Author):

The authors have addressed my comments with additional bioinformatic analysis and novel experimental data. The results of these experiments overall support the original hypothesis.

I have only a minor comment with regard to the bioinformatic analysis:

it seems strange that the TCGA-CRC cohort contains only few number of cases as indicated in the response Figure 8A. The authors should indicate which specific TCGA dataset is used as there are several. I recommend to reanalyze with the TCGA PanCancer dataset. Furthermore, the analysis should be clearly described in the methods section.

Overall, the manuscript has been significantly improved and once the minor comment has been addressed, it is suitable for publication.

Response: Thanks a lot for the reviewer's constructive comments. We analyzed the TCGA Colorectal Adenocarcinoma proteomic data downloaded from the cBioportal website (<https://www.cbioportal.org/datasets>). In that dataset, two proteomic methods were used including protein mass spectrometry and RPPA (Revers Phase Protein microArray). As to the RPPA result, no EIF3H or HAX1 signal were captured in that data, thus it was not further analyzed. As to the mass spectrometry result (https://www.cbioportal.org/study/summary?id=coadread_tega), only 90 patients' samples were recorded in the dataset, and we further excluded the samples that either EIF3H or HAX1 data was missing, which finally came out to be 38 patients' samples left. That is why TCGA-CRC cohort contains only few number of cases (Response Figure a).

As for TCGA PanCancer dataset, from the cBioportal website, only three types of TCGA PanCancer, including breast cancer (n=105), colorectal cancer (n=90) and ovarian cancer (n=121), have the protein mass spectrometry data. We further analyzed the correlation of EIF3H and HAX1 expression in breast cancer and ovarian cancer. However, there was no correlation between EIF3H expression and HAX1 expression in TCGA Breast and Ovarian Cancer (Response Figure b and c).

For the methods section, we added it to the manuscript as following: Bioinformatics analysis: TCGA PanCancer Proteomic data, including breast cancer (n=105), colorectal cancer (n=90) and ovarian cancer (n=121), were downloaded from the cBioportal dataset (<https://www.cbioportal.org/datasets>). Analyzed the mass spectrometry data and the protein

correlation with Z-score using Pearson's correlation (Graphpad Prism 7). Mass spectrometry data of CPTAC colon cancer cohort were analyzed by online tool (<https://cprosite.ccr.cancer.gov/>). The mNRA Correlation data were also derived from online tool (<https://cprosite.ccr.cancer.gov/>).

Response Figure a-c. Pearson's correlation analysis of EIF3H and HAX1 protein expression in tumors from indicated TCGA cohort. CRC, Colorectal Adenocarcinoma; BRCA, Breast Cancer; OV, Ovarian Serous Cystadenocarcinoma.